# The tropical route of QBO teleconnections in a climate model

Jorge L. García-Franco[1], Lesley J. Gray[1,2], Scott Osprey[1,2], Robin Chadwick[3,4], and Zane Martin[5]

[1]Atmospheric, Oceanic and Planetary Physics, University of Oxford, Oxford, UK
[2]National Centre for Atmospheric Science, UK
[3]Met Office Hadley Centre, Exeter, UK
[4]Global Systems Institute, Department of Mathematics, University of Exeter, Exeter, UK
[5]Department of Atmospheric Science, Colorado State University, Fort Collins, CO

**Correspondence:** Jorge L. García-Franco (jorge.garcia-franco@physics.ox.ac.uk)

**Abstract.** The influence of the quasi-biennial oscillation (QBO) on tropical climate is demonstrated using 500-yr pre-industrial control simulations from the Met Office Hadley Centre model. Robust precipitation responses to the phase of the QBO are diagnosed in the model, which show zonally asymmetric patterns that resemble the El Niño-Southern Oscillation (ENSO) impacts. These patterns are found because the frequency of ENSO events for each QBO phase is significantly different in these simulations, with more El Niño events found under the westerly phase of the QBO (QBOW) and more La Niña events for the easterly phase (QBOE). The QBO-ENSO relationship is non-stationary and subject to decadal variability in both models and observations. In addition, regression analysis shows that there is a QBO signal in precipitation that is independent of ENSO. No evidence is found to suggest that these QBO-ENSO relationships are caused by ENSO modulating the QBO in the simulations. A relationship between the QBO and a dipole of precipitation in the Indian Ocean is also found in models and observations in boreal fall, characterized by a wetter western Indian Ocean and drier conditions in the eastern part for QBOW and the opposite under QBOE conditions. The Walker circulation is significantly weaker during QBOW compared to QBOE, which could explain the observed and simulated zonally asymmetric precipitation responses at equatorial latitudes as well as the more frequent El Niño events during QBOW. Further work, including targeted model experiments, is required to better understand the mechanisms causing these relationships between the QBO and tropical convection.

## 1 Introduction

Long-distance effects or teleconnections associated with the stratospheric quasi-biennial oscillation (QBO) have been well documented in the subtropics and extratropics, including for example: the stratospheric polar vortex (Holton and Tan, 1980; Anstey and Shepherd, 2014; Domeisen et al., 2019; Lu et al., 2020), the subtropical jets (Garfinkel and Hartmann, 2011b; Hansen et al., 2016; Ma et al., 2021) and the North Atlantic Oscillation (Hansen et al., 2016; Gray et al., 2018; Andrews et al., 2019b). Observational and modelling evidence suggests that there is also a tropical route of influence of the QBO through impacts on: monsoons (Giorgetta et al., 1999; Claud and Terray, 2007; Liess and Geller, 2012), the Intertropical Convergence Zone (ITCZ) (Gray et al., 2018), tropical sea-surface temperatures (SSTs) (Garfinkel and Hartmann, 2011a; Huang et al., 2012), tropical high clouds (Liess and Geller, 2012; Peña-Ortiz et al., 2019), tropical cyclones (Ho et al., 2009; Jaramillo et al.,

2021), and the Madden-Julian Oscillation (MJO) (Son et al., 2017; Wang et al., 2019; Martin et al., 2021b). For recent reviews
on stratosphere-troposphere coupling in the tropics, see Haynes et al. (2021) and Hitchman et al. (2021).

The tropical route of QBO teleconnections remains less well understood than other routes for various reasons. One reason is that the observational record is too short to diagnose robust differences between the two QBO phases in a 30-40-yr long dataset, as variability in the tropics on QBO time-scales is dominated by El Niño-Southern Oscillation (ENSO) (Liess and Geller, 2012; Seo et al., 2013; Gray et al., 2018). Similarly, the modulation of the location and strength of tropical convection
by ENSO events can influence the characteristics of the QBO (Taguchi, 2010; Schirber, 2015; Geller et al., 2016; Christiansen et al., 2016; Serva et al., 2020), which makes it difficult to separate the cause and effects of ENSO and the QBO.

However, multiple lines of evidence suggest that there is a modulation of several features of tropical climate by the QBO. In observations, surface impacts of the QBO over monsoon regions have been diagnosed in satellite-derived fields such as cloud height, occurrence and out-going longwave radiation (Collimore et al., 2003; Liess and Geller, 2012), as well as in surface
precipitation (Seo et al., 2013; Gray et al., 2018). The observational evidence shows zonally asymmetric impacts – indicating that the QBO influence depends on longitude. A proposed mechanism suggests a QBO modulation of the Walker circulation (Collimore et al., 2003; Liess and Geller, 2012). Additionally, analyses of observations and forecast models have found that the MJO is stronger and more predictable under the easterly phase in the lower stratosphere (Son et al., 2017; Lim et al., 2019; Klotzbach et al., 2019; Martin et al., 2020, 2021b).

Several of the observed relationships between the QBO and tropical convective phenomena appear to be non-stationary or intermittent. For example, Gray (1984) found evidence for a QBO modulation of Atlantic tropical cyclones but this link seemingly disappears after 1990 (Camargo and Sobel, 2010). A second example is the QBO-ENSO relationship; during the 1953-1980 period, La Niña (LN) events were more frequent under the westerly phase of the QBO (QBOW) and El Niño (EN) events during the easterly phase (QBOE) suggesting an anti-correlation of QBO-ENSO indices (Garfinkel and Hartmann, 2007;
Hu et al., 2012; Domeisen et al., 2019). However, the relationship between the QBO and ENSO indices has become positive since 1985 (Taguchi, 2010; Liess and Geller, 2012). The observed MJO-QBO relationship has also only appeared after 1980 (Klotzbach et al., 2019). Intercadal variations have also been observed in the QBO-Walker circulation relationship, which since 1980 has been characterized by a stronger (weaker) Walker circulation under QBOE (QBOW) (Hu et al., 2012; Hitchman et al., 2021).

Modelling studies have also investigated tropical surface impacts associated with the QBO. For example, Giorgetta et al. (1999) found that boreal summer monsoon regions exhibit a significant response in cloudiness to the QBO winds in a General Circulation Model (GCM). In a cloud-resolving model, Nie and Sobel (2015) found that the influence of the QBO may depend on the strength of convection and SST forcing, suggesting a non-linear effect of the QBO on the convective processes.

A relatively smaller number of studies have analysed tropical QBO teleconnections in global climate models. For example,
Rao et al. (2020) analysed the precipitation response to the QBO in models from the Coupled Model Intercomparison Project phase 6 (CMIP6) cohort and found large model disagreement in the sign and pattern of the precipitation response. In contrast, Serva et al. (2022) analysed simulations from the QBO-initiative (QBOi) (Butchart et al., 2018) project and found robust

responses across models in the East Pacific ITCZ that are similar to the observed pattern (Gray et al., 2018). Kim et al. (2020) found that the CMIP6 models cannot reproduce the observed QBO-MJO relationship.

Moreover, the physical mechanisms through which the QBO could influence tropical climate are also not well understood. The influence of the QBO over the temperature and vertical wind shear near the tropopause layer (Tegtmeier et al., 2020; Martin et al., 2021c) has been hypothesized to affect tropical deep convection. For example, early studies (Gray, 1984; Collimore et al., 2003) argue that the QBO induces changes to the vertical wind shear or static stability in the upper-troposphere lower-stratosphere (UTLS) that can modify the depth of convection at equatorial latitudes. However, other studies suggest that the

surface impact of the QBO may be a function of both the UTLS temperature changes and the tropospheric convective forcing (Nie and Sobel, 2015).

In short, observational uncertainty limits the confidence in the diagnosis of impacts from the QBO to the tropical troposphere.

Long integrations of GCMs could provide better statistics and understanding of the surface impact of the QBO in the tropics. This paper aims to address these shortcomings by investigating the tropical route of QBO influence using long integrations of

the UK Met Office Hadley Centre (MOHC) Unified Model submitted to CMIP6. The model extends to the mesosphere and includes a self-generated QBO via a non-orographic gravity wave scheme that compares well with the observed QBO (Richter et al., 2020). The CMIP6 pre-industrial control (piControl) experiments with constant 1850 external forcing are examined to exploit their length (500-yr) and the resulting statistical robustness of the diagnosed QBO impacts.

The main purpose of this investigation is to diagnose whether there are any large-scale QBO impacts on the tropical tropo-

sphere within a long integration of a state-of-the-art GCM. QBO-MJO connections are excluded from this study as they have already been explored and found largely absent in the MOHC models (Kim et al., 2020).

The paper is structured as follows. Section 2 describes the simulations together with the observational and reanalysis data that are employed for comparison and verification, as well as the composite and regression techniques employed in the study. Section 3 examines evidence for QBO signals in a variety of tropical climate indicators, including precipitation, the ITCZ,

monsoons, ENSO, the Walker circulation and the Indian Ocean Dipole (IOD). The final section provides a summary and conclusions of the main findings.

## 2    Methods and data

### 2.1    Observations and reanalysis

The gridded precipitation datasets used in this study are the $1°$ resolution Global Precipitation Climatology Project (GPCP)

v2.3 (Adler et al., 2003) dataset and the Global Precipitation Climatology Centre (GPCC) dataset version 6 at $0.5°$ resolution (Becker et al., 2011; Schneider et al., 2011); in both cases available as monthly-means. GPCP is a merged product of satellite and land rain-gauge observations and provides coverage over land and ocean, whereas GPCC uses a large network of surface station data going back to the early 1900s and has a higher horizontal resolution but does not provide data over oceanic regions (Becker et al., 2013; Adler et al., 2018). Zonal winds at the 70 hPa level from the Freie Universität Berlin (FUB) radiosonde

dataset and from a long reconstruction (1930-2021 in this study) from sea-level pressure data (hereafter B07 Brönnimann et al., 2007) are also used to diagnose the QBO (see section 2.3).

    For the other diagnostics, including zonal wind, vertical velocity and convective precipitation, we use the European Centre for Medium-Range Weather Forecasts (ECMWF) ERA5 Reanalyses (Hersbach et al., 2020) downloaded at the $0.75° \times 0.75°$ resolution. In most instances, monthly-averaged ERA5 data was used, but for the computation of diagnostic quantities such as

the zonal streamfunction, hourly data was downloaded and then averaged to daily-mean values. The GPCP and ERA5 datasets span the 1979-2021 period whereas the period 1953-2019 is used for GPCC.

## 2.2   CMIP6 data

    The MOHC submitted three simulations for the piControl experiment of CMIP6 using two models: HadGEM3 and UKESM1. The HadGEM3 model is the core physical climate model and UKESM1 is an Earth System Model extension, with additional

treatment of aspects of e.g. land surface, ocean and sea-ice processes as well as improved chemical processes (Kuhlbrodt et al., 2018; Williams et al., 2018; Sellar et al., 2019). The three piControl experiments analysed in this study are: HadGEM3 GC3.1 at N96 and N216 horizontal resolutions (hereafter referred to as GC3 N96-pi and GC3 N216-pi) and UKESM at N96 horizontal resolution (hereafter referred to as UKESM N96-pi). The $N$ in N96 refers to the maximum number of zonal 2 grid-point waves that can be represented by the model (Walters et al., 2019) at that resolution, so that the N96 and N216 atmospheric resolutions

at the midlatitudes are $1.875° \times 1.25°$ and $0.83° \times 0.56°$, respectively, whereas their oceanic resolutions using the NEMO model are $1°$ (ORCA1) and $0.25°$ (ORCA025) (Williams et al., 2018), respectively.

    The 500 years available for GC3 N216-pi are used and although more data exists for UKESM N96-pi and GC3 N96-pi, we use 500-yrs of these simulations for statistical consistency. The three simulations have the same experimental design with constant year-1850 external forcing, further detail about the MOHC piControl experiments can be found in Menary et al. (2018)

and about the UKESM1 model in Sellar et al. (2019). The majority of diagnostics are shown for GC3 N216-pi simulation and comparisons with the other two simulations are noted where appropriate.

    The equatorial climate of GC3 N216-pi captures tropical dynamical processes including mean and extreme precipitation reasonably well (García-Franco et al., 2020; Abdelmoaty et al., 2021), as this configuration is amongst the best compared to other CMIP5/CMIP6 models, e.g., in tropical extreme precipitation (Abdelmoaty et al., 2021) and the annual cycle of equatorial

Atlantic SSTs and low-level winds (Richter and Tokinaga, 2020). However, notable biases of the MOHC models include the southward bias of the Atlantic ITCZ linked to the dry Amazon bias and a wet bias over the East Pacific ITCZ (García-Franco et al., 2020).

    Furthermore, the MOHC models have improved their representation of ENSO characteristics, e.g., Lee et al. (2021) finds that HadGEM3 configurations represent the pattern, seasonal cycle, amplitude and lifecyle of ENSO better than the CMIP6

multi-model mean. These results agree with other studies that indicate the GC3 N216-pi configuration reasonably simulates the seasonal phase-locking and the spectral power of ENSO (Menary et al., 2018; Richter and Tokinaga, 2020; Liu et al., 2021). However, this configuration underepresents extreme ENSO events and the observed "ENSO diversity" in the SST patterns (García-Franco et al., 2020).

In the stratosphere, this and previous configurations of the model reasonably simulate the QBO (Schenzinger et al., 2017; Richter et al., 2020; Bushell et al., 2020). The model QBO is driven by both resolved and parametrised non-orographic gravity waves (Scaife et al., 2002; Walters et al., 2019) and is tied to total precipitation sources in the tropics (Bushell et al., 2015). However, the atmosphere model configuration used in this study underestimates the amplitude of the QBO in the lower strato-sphere (60-90 hPa), with the maximum bias found at 70 hPa of 5 m s$^{-1}$, and the power spectrum of QBO periods shows more power at longer periods (32-36 months) than observations (Bushell et al., 2020).

## 2.3 Indices

The index used to characterise the QBO for ERA5, the FUB and the simulations is the monthly-mean zonal-mean zonal winds at 70 hPa averaged between 5°S-5°N, which is well suited to diagnose impacts in the tropical tropopause region (Huesmann and Hitchman, 2001; Gray et al., 2018; Hitchman et al., 2021; Serva et al., 2022). The QBO phase is defined using a threshold of 2 m s$^{-1}$ (Garfinkel and Hartmann, 2010), so the transition months where the QBO winds fall within the range $\pm$ 2 m s$^{-1}$ are excluded. The reconstruction from B07 uses the 90 hPa winds and a threshold of $\pm$ 2 m s$^{-1}$. The amplitude and descent rates of the QBO are calculated using the deseasonalized zonal mean zonal wind averaged over the stated latitudes between 10 and 70 hPa. The amplitude ($A$) of the QBO is defined using the first and second principal components (PCs) following an empirical orthogonal function (EOF) decomposition of the 10-70 hPa wind time-series (Serva et al., 2020) as $A = \sqrt{PC_1^2 + PC_2^2}$. The descent rates are calculated following Schenzinger et al. (2017) for descending westerly and easterly phases individually by finding the level of the zero wind line ($u = 0$) for each month computing the height difference between consecutive months. These definitions of the amplitude and descent rates were chosen to evaluate the influence of ENSO on the whole profile of the QBO and not just one single level.

The EN3.4 SST index is used to characterise ENSO by area-averaging the box within [5°S-5°N] and [190°E-240°E]. A 5-month running-mean of the index is calculated and a threshold of $\pm$0.5 K used to define positive (EN) and negative (LN) events. Neutral months (NN) are defined where the magnitude of the EN3.4 index is smaller than $\pm$0.5 K. For the Indian Ocean Dipole (IOD), an index to characterise the zonal gradient in convective precipitation in the Indian Ocean (convective IOD Index) was defined as the difference of the deseasonalized area-averaged convective precipitation between the western [50-70°E] and eastern [80-100°E] equatorial [10°S-10°N] Indian Ocean, which is in a similar region as the standard SST IOD index (Wang and Wang, 2014). This index was computed using the convective precipitation from the models and ERA5, and IOD events were defined using a 1 standard deviation threshold.

## 2.4 Analysis techniques

Composite analysis is the primary technique used in the study. Annual-mean and seasonal-mean composites were derived by computing weighted averages to account for differences in the number of samples from each month and avoid a possible seasonal effect due to QBO or ENSO phase-locking, so all months contribute equally to a seasonal or annual-mean composite.

The length of the experiments is such that the number of total El Niño and La Niña months for GC3 N216-pi were 1700 and 1600, respectively, whereas 2400 months were classified as QBOW and 1800 as QBOE. Moreover, EN months found under

QBOW were 626 compared to 392 under QBOE for the simulation, whereas in the observed 1979-2020 period using HadSST SSTs and the ERA5 QBO index, 65 QBOW EN months and 45 QBOE EN months were diagnosed. The ratio of QBOW-EN and QBOE-EN months to the total number of available months is slightly lower in the models (0.10 and 0.06, respectively) compared to ERA5 1979-2020 (0.11 and 0.08, respectively) because in the models fewer months satisfy our threshold for each QBO phase at 70 hPa, due to the low amplitude bias of the QBO in the lower stratosphere in GCMs. (Schenzinger et al., 2017; Bushell et al., 2020).

In addition to composite analysis, multi-linear regression analysis is also employed to explore the impact of one or more of the indices. Previous studies have shown that the regression can separate candidate mechanisms or indices (Gray et al., 2018; Misios et al., 2019; Rao et al., 2020), for example, by removing the influence of ENSO. Details of the regression analysis technique are provided in the appendix A1.

The statistical significance of the observed composite differences is estimated using a randomised resampling ("bootstrapping with replacement") method that generates a distribution of differences constructed from randomly sampling the observed period. The significance level is then interpreted as QBO W-E differences that are outside the 95% of the distribution of random differences. The significance in the simulations is estimated using a Welch two-sided t-test. When calculating correlations, random samples are drawn from the entire simulation and the process is repeated 10,000 times to evaluate the likelihood of obtaining a significant relationship by chance.

## 3 Results

The tropical precipitation response to the QBO phase is analysed first in the annual-mean and then by season (section 3.1). The potential for aliasing with the ENSO signal is investigated (section 3.2) and QBO-ENSO interactions are further explored (section 3.3), as well as QBO interactions with the Indian Ocean dipole (IOD). Finally, interactions between the QBO and the ITCZ, monsoons and the Walker circulation are identified and discussed in section 3.4.

### 3.1 Precipitation

QBO composite differences in annual mean precipitation (QBO-W minus QBO-E) are shown in Figure 1 from the gridded GPCP observational dataset and from all three model simulations. In the observations the QBO signals are largest and statistically significant in the tropical Pacific, equatorial Atlantic and Indian Oceans, in good agreement with previous analyses (Liess and Geller, 2012; Gray et al., 2018). The three simulations agree reasonably well with the GPCP distributions and amplitudes, particularly in the Pacific and Indian Oceans. Positive differences of up to 1.2 mm day$^{-1}$ are found in the equatorial Central Pacific and the Indian Ocean and negative differences of up to 0.6 mm day$^{-1}$ in the off-equatorial North Pacific, although the differences are smaller in the simulations than observed.

In the tropical Atlantic, however, there is an indication of a weak but significant signal in the observations near the ITCZ but the models show a signal of the opposite sign in this region (or the absence of a signal in the case of GC3 N96-pi). This disagreement with observations may be due to the biased southward position of the Atlantic ITCZ in the model which is more

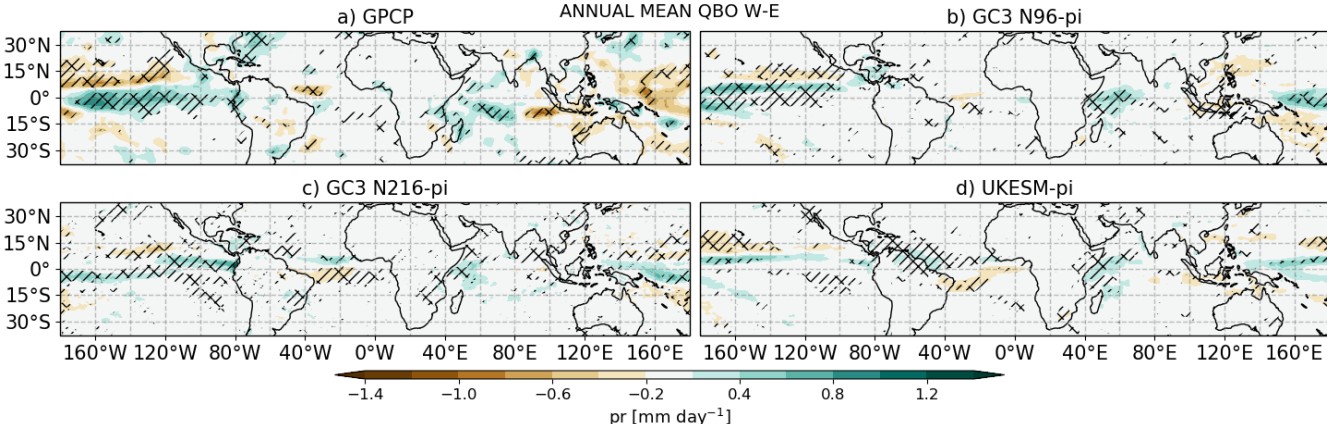

**Figure 1.** QBO-W minus QBO-E composite differences in annual-mean precipitation [mm day$^{-1}$] from (a) GPCP, (b) GC3 N96-pi, (c) GC3 N216-pi and (d) UKESM N96-pi. Hatching denotes statistically significant differences at the 95% confidence level.

pronounced in December-January-February (DJF) (García-Franco et al., 2020). This ITCZ response is investigated further in section 3.4.

The QBO signal in precipitation is found to be strongly dependent on the seasonal cycle in both models and observations. Figure 2 shows a comparison of the GPCP dataset and GC3 N216-pi for individual seasons (see Figure S1 for the other models). The positive equatorial Pacific signal in the GPCP dataset, which resembles an El Niño anomaly (Dommenget et al., 2013; Capotondi et al., 2015), is particularly strong and statistically significant in DJF (Fig. 2a). A similar pattern is present in MAM (Figure 2c) but with no statistical significance. In GC3 N216-pi, the QBO signal in the Pacific is significant in all seasons but is generally weaker than observations, likely due to the greater number of years in the simulation. The Pacific response is shifted slightly south compared to GPCP and is strongest in MAM instead of DJF (Fig. 2c).

In the Atlantic, the QBO signal in the ITCZ region is more evident in the individual seasons. In DJF there is reasonable similarity between the model and observations but by MAM the opposite sign of the model compared to GPCP becomes evident, even though the ITCZ bias in the model is stronger in DJF than in MAM (García-Franco et al., 2020). In addition, all models and GPCP indicate that the Caribbean Sea is wetter in JJA during QBO-W than in QBO-E (see Figs. 2 and S1). In the Indian Ocean, the observations and all models show relatively large and significant differences in SON, (Fig. 2e-f), characterized by a dipole of wet anomalies to the west and dry anomalies to the east. The dipole anomalies suggest a possible QBO influence on the IOD, which is characterized by a zonal gradient of SSTs and convective activity in the Indian Ocean that is specially prominent in SON (Saji et al., 1999; Deser et al., 2010; McKenna et al., 2020). This possibility is explored further in section 3.3 below.

In summary, the GPCP total precipitation composite analyses show a zonally asymmetric QBO signal primarily in the ITCZ regions over the oceans, consistent with previous studies (Liess and Geller, 2012; Gray et al., 2018). The modelled QBO response patterns are in good agreement with the GPCP analysis albeit with small shifts in the timing of the maximum response

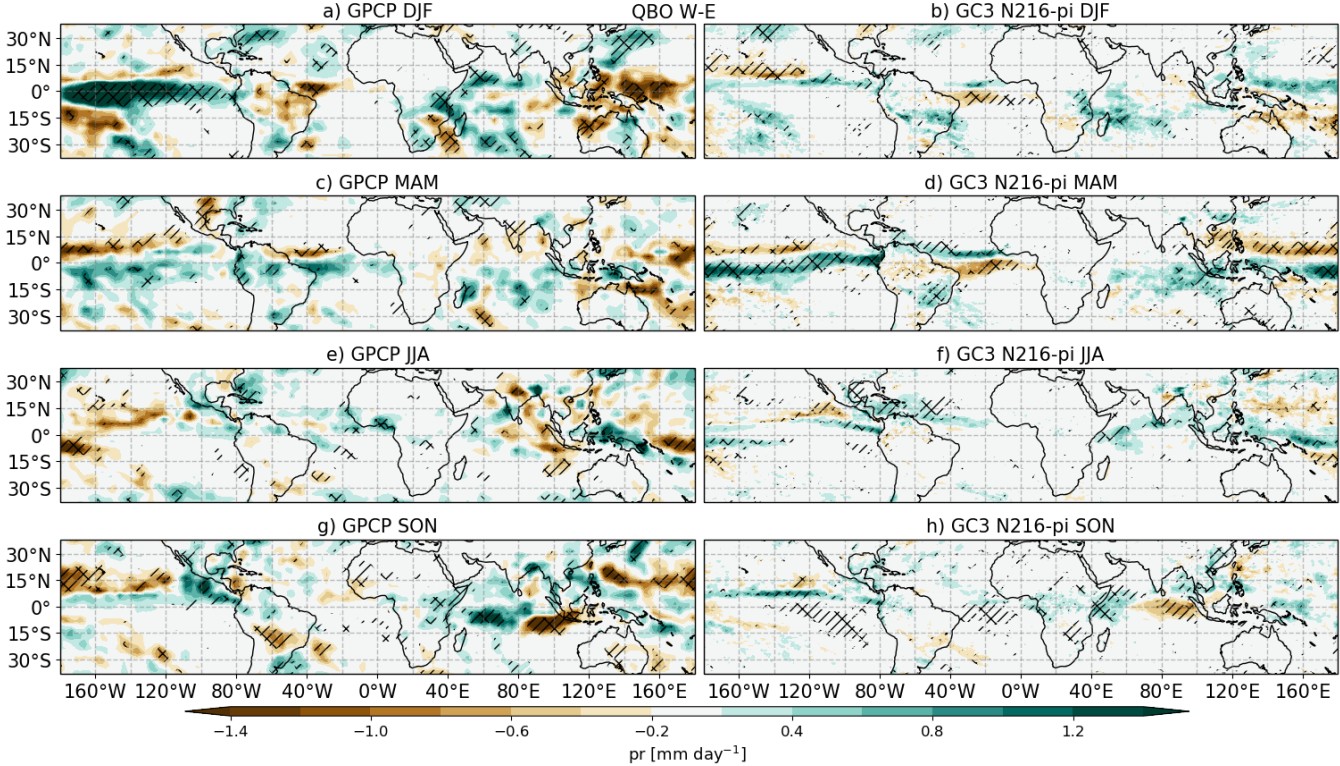

**Figure 2.** As in Figure 1, but showing seasonal-mean QBO composite from (left) GPCP and (right) GC3 N216-pi for (a,b) December-January-February (DJF), (c,d) March-April-May (MAM), (e,f) June-July-August (JJA) and (g,h) September-October-November (SON) from top to bottom.

in the Pacific and an overall weaker response than observations. The significant response to the QBO in these long simulations (500 years) suggests that the analysis of the modelled QBO signals may help to understand the mechanisms that give rise to the QBO signals at the surface. However, the QBO signals from both model and observational analyses show strong similarities to well-known response patterns for ENSO and the IOD. Before further investigating the QBO surface impacts, these tropical interactions are investigated in the following sections.

## 3.2 Potential aliasing of QBO and ENSO signals

The tropical SSTs and the EN3.4 index differences for each QBO phase are investigated in this section to understand the precipitation patterns found in Figures 1 and 2. The SST response (QBO W-E) closely follows the precipitation patterns for observations and models (Figs. 3, S2 and S3), with warmer SSTs found in regions with increased precipitation. In DJF, warmer SSTs in the equatorial Pacific and western Indian Oceans are observed for QBOW compared to QBOE. In particular, the pattern in the HadSST dataset since 1979 resembles an East Pacific (or 'standard') El Niño, whereas the simulated anomalies in DJF (see also Fig. S3) are weaker and resemble a central Pacific El Niño (Capotondi et al., 2015).

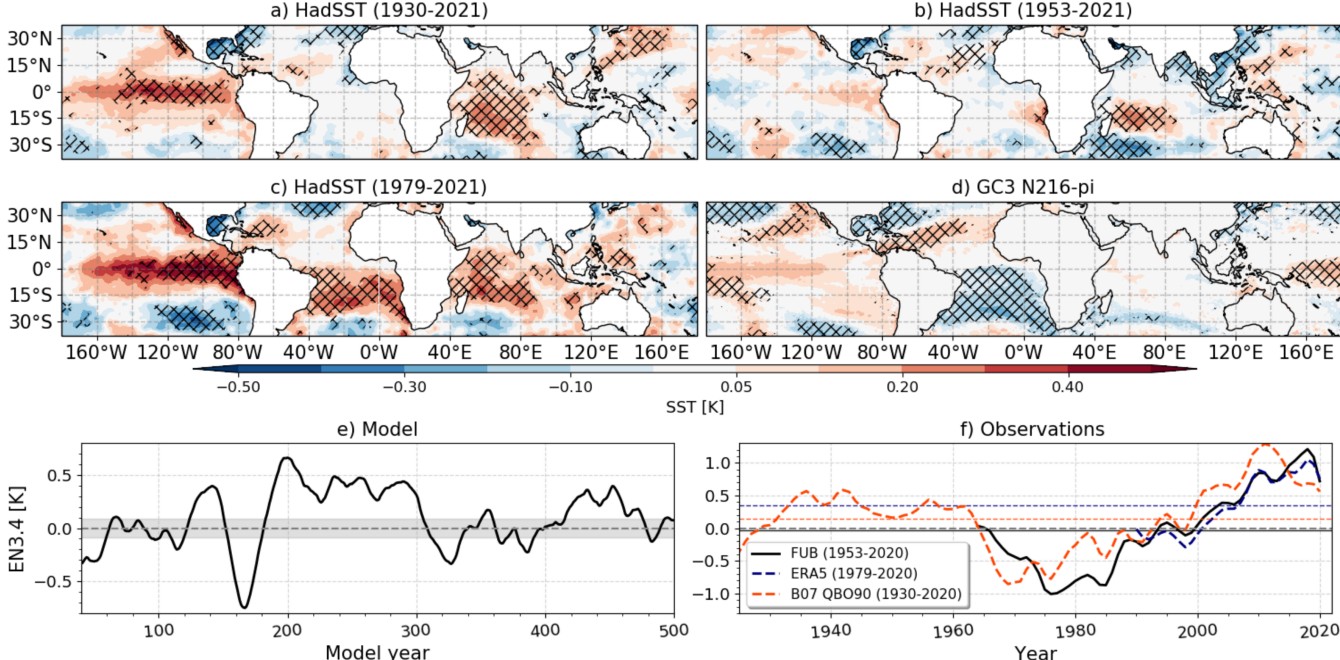

**Figure 3.** (a-d) SST differences (QBO W-E) in DJF for (a-c) HadSST and (d) GC3 N216-pi. In (a-c) the HadSST differences are obtained by using (a) the B07 reconstruction (1930-2021), (b) the FUB dataset and (c) ERA5. (e-f) show time-series of EN3.4 [K] differences (QBO W-E) in (e) GC3 N216-pi and (f) HadSST using the different products for the QBO. In (e) the time-series are constructed by computing the W-E differences in NDJFM in sliding windows of 30 year periods and in (f) by computing the 12-yr sliding average. The shading in (e) indicates the first and third quartile of a distribution of differences found by randomized resampling and the horizontal lines indicate in (f) the mean EN3.4 for each dataset.

Previous studies have noted (Garfinkel and Hartmann, 2007; Domeisen et al., 2019) that the QBO-ENSO relationship in observations has changed since 1979. Figure 3 confirms that the Pacific SST response to the QBO has changed over different decades, with an anti-correlation relationship in 1960-1980 and a positive relationship emerging from 1985 to present. In the period prior to the radiosonde era (1930-1960) the relationship was also positive, according to the reconstructed index (B07).

The model simulations also exhibit decadal variability in the ENSO-QBO relationship (Fig. 3e). Even though the QBO-ENSO relationship is positive for the most part of the simulation, some 30-50 yr periods exhibit a negative relationship and other periods exhibit a difference that is comparable and even higher than observed. The link between the variability of the QBO-ENSO relationship in the model and the Atlantic Multidecadal variability (Sutton and Hodson, 2005) or the Pacific Decadal Oscillation (Mantua et al., 1997) indices was investigated (not shown) and no significant relationship or influence was found. In observations, it is plausible that the apparent QBO-ENSO relationship occurs by chance because of the relatively short data record, or that it is influenced by external forcing and internal variability causing the QBO-ENSO relationship to change sign. In the simulations, however, the length of the dataset rules out the possibility of aliasing by chance, and there is no

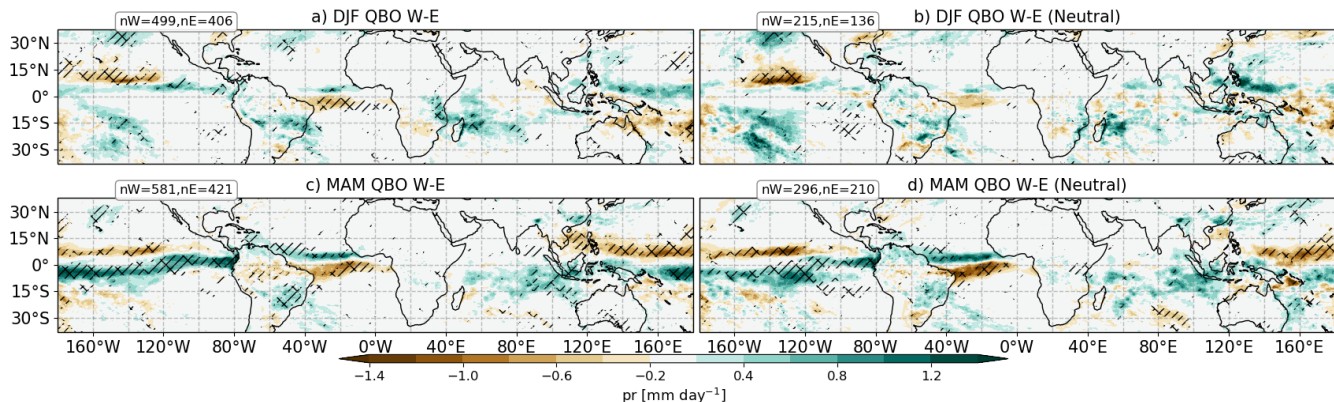

**Figure 4.** Composite QBO W-E differences of total precipitation in GC3 N216-pi in (a, b) DJF and (c, d) MAM for (a, c) all the events and (b, d) Neutral ENSO conditions only. The sample size of each composite is noted in the top left corner of each panel. Statistically significant differences to the 95% confidence level are shown through the hatching.

variation of external forcing. In summary, both the observations and the model simulation suggest that there is a relationship
between the QBO and ENSO that requires further investigation.

As an initial investigation of the possibility of aliasing between the QBO and ENSO signals, Figures 4a,b shows the DJF
QBOW minus QBOE composite differences of total precipitation from the GC3 N216-pi simulation using all years (as in figure
2) compared with using only those years identified as 'ENSO-neutral'. Although the sample size is substantially reduced in
the latter (see figure for the number of months in each QBO composite) the sample size is still large. The response patterns are
similar in each plot e.g., the drier patch north of the equator over the Eastern Pacific or the wet anomaly over Madagascar. This
result suggests that there is a QBO signal that is unlikely to be the result of a sampling bias that favours one particular phase of
ENSO. However, the equatorial Atlantic and Pacific MAM responses are stronger when ENSO events are included.

An alternative approach to investigate the possibility of aliasing of the QBO/ENSO signals is to use a multi-linear regression
technique (see section 2) in which the signal is analysed for both QBO and ENSO simultaneously. Here, we switch to analysing
convective precipitation to better investigate the possible influence of the QBO on deep tropical convection.

Figures 5a,b show results from a simple linear regression analysis of the monthly-averaged time-series of GC3 N216-pi
total precipitation in which only the 70 hPa QBO index was employed. Figure 5a includes all available years while Figure 5b
includes only neutral ENSO years. The results are very similar to the annual-mean composite differences in total precipitation
(Fig. 1), with increased convective precipitation over the equatorial Pacific when the zonal winds at 70 hPa are positive.

Figures 5c,d show the ENSO and QBO signals when the EN3.4 index is included as well as the QBO index. The ENSO
response is clearly evident, highly statistically significant and compares well with the well-known patterns obtained from
observations. The amplitude of the ENSO signal is also much larger than the QBO signal. Nevertheless, the QBO signal
remains intact and all of the main features are still significant (Fig. 5c). For example, the positive regression coefficients that
suggest a northward shift of the Atlantic ITCZ and the wetter Caribbean Sea and western Indian Ocean in the simple regression

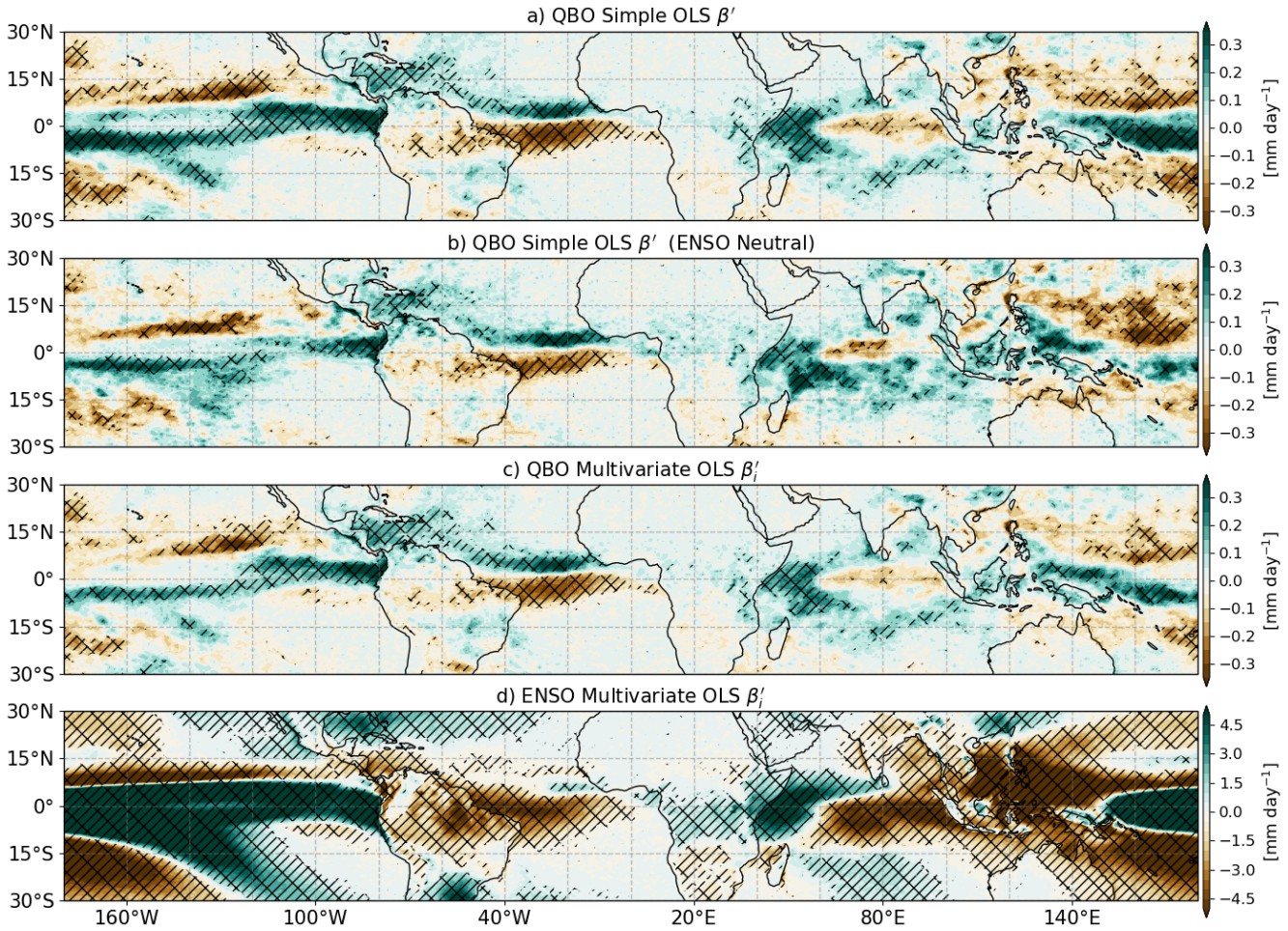

**Figure 5.** Annual-mean regression model results in GC3 N216-pi for convective precipitation. (a, b) Rescaled regression coefficients ($\beta_i'$) from a simple ordinary least-squares (OLS) regression model with the QBO index for (a) all months and (b) ENSO-Neutral months only. (c, d) show the regression coefficients resulting from a multivariate regression model using the ENSO and QBO indices for the (c) QBO and (d) predictors. In all cases, the regression coefficients are rescaled by multiplying the regression coefficients with the ratio of maximum amplitude and standard deviation of the QBO or ENSO indices.

model are still found in the multivariate regression analysis. A similar analysis of tropical SSTs (Fig. S4) shows a QBO signal in SSTs that is separate from the effect of ENSO and agrees with the results of the composite analysis (Fig. 3).

These results suggest that the modelled QBO signal in total precipitation does not arise due to a simple aliasing of the signal with ENSO. However, the multi-linear regression technique assumes that the QBO and ENSO indices are linearly independent and that their responses add together linearly. The similarity of the two responses suggests that this is probably not the case and there may be substantial non-linear interaction between the two phenomena (e.g. Gray et al., 1992). Nevertheless, the QBO

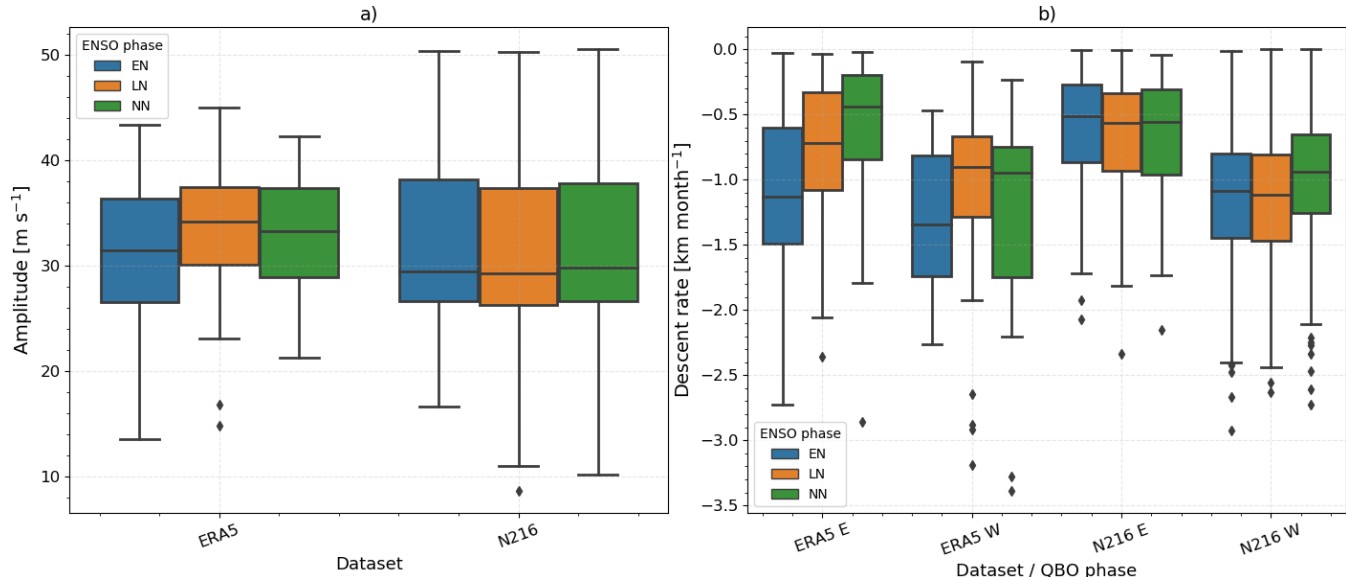

**Figure 6.** Box plots of the median (horizontal line), first and third quartile (boxes) and minimum and maximum (delimited by the whiskers) of the distribution of QBO (a) amplitudes [m s$^{-1}$] and (b) descent rates [km month$^{-1}$] separated by dataset (ERA5 and GC3 N216-pi) and ENSO phase. NN stands for Neutral ENSO. In (b) descent rates are shown for both descending easterly (E) and westerly (W) phases following Schenzinger et al. (2017).

signal remains even when only neutral-ENSO years are included in the analysis, suggesting that the QBO has a real influence on the surface precipitation.

### 3.3 Interaction of the QBO with ENSO and IOD

To further explore the interaction of the QBO and ENSO, we first investigate whether aspects of the QBO are influenced by
ENSO events. This relationship could be a real possibility since the intrinsic mechanism of the QBO involves tropical waves that are generated within the troposphere. Schirber (2015) found in a GCM that under El Niño conditions tropospheric wave activity increases and accelerates the downward propagation speed of the QBO westerly phase. However, the analysis by Serva et al. (2020) shows that only models with relatively high horizontal resolution can reproduce the observed ENSO effects on the QBO amplitude while several models, including the UM, show no impact of ENSO on the QBO.
For that reason, we analyse several characteristics of the QBO and their dependence on ENSO phase, namely the descent rate and the amplitude of the QBO (see section 2.3 for details of how these metrics are defined). The results are summarised in Figure 6 for the ERA5 reanalysis dataset (41 years; 1979 – 2020) and for the GC3 N216-pi simulation (500 years). In ERA5, the well-known faster descent rates during the westerly phase than in the easterly phase is evident and agrees well with studies of longer datasets such as the Berlin radiosonde data (Schenzinger et al., 2017). Also, the ERA5 QBO descent rates and the

**Table 1.** ENSO and IOD events frequency (month month$^{-1}$), e.g., (# months EN) / (# months W). For each mean value the standard deviation of the probability density distribution (PDF) is shown found by boostrapping with replacement. Note that the ENSO frequencies for ERA5 (1979-2020), FUB (1953-2020) and B07 (1930-2020) are obtained using HadSST data. Results in **bold** indicate that the event frequency PDF for QBOW is significantly different to QBOE to the 95% confidence level according to the KS test.

| Dataset | QBO phase | El Niño | La Niña | IOD+ | IOD- |
|---------|-----------|---------|---------|------|------|
| FUB | W | 0.22±0.02 | 0.28±0.02 | - | - |
| FUB | E | 0.28±0.03 | 0.23±0.03 | - | - |
| B07 | W | 0.25±0.02 | 0.23±0.03 | - | - |
| B07 | E | 0.22±0.03 | 0.27±0.03 | - | - |
| ERA5 | W | 0.28±0.02 | 0.27±0.02 | 0.17±0.03 | 0.11±0.02 |
| ERA5 | E | 0.24±0.02 | 0.27±0.03 | 0.12±0.01 | 0.16±0.03 |
| GC3 N216-pi | W | **0.27±0.1** | **0.19±0.05** | **0.17±0.03** | **0.11±0.02** |
| GC3 N216-pi | E | **0.24±0.08** | **0.26±0.07** | **0.12±0.03** | **0.15±0.03** |
| GC3 N96-pi | W | **0.33±0.09** | **0.21±0.06** | **0.18±0.04** | **0.12±0.03** |
| GC3 N96-pi | E | **0.26±0.09** | **0.27±0.07** | **0.13±0.04** | **0.14±0.03** |
| UKESM-pi | W | **0.30±0.08** | **0.24±0.06** | **0.16±0.04** | **0.12±0.04** |
| UKESM-pi | E | **0.27±0.10** | **0.28±0.09** | **0.13±0.04** | **0.15±0.04** |

amplitude both depend on the phase of ENSO. A higher amplitude and slower descent rates are observed during La Niña phases and weaker amplitudes and faster descent rates during El Niño, in agreement with Geller et al. (2016).

In the model, the descent rates are also faster for the westerly than the easterly QBO phase, as observed, but the relationship between the QBO characteristics and ENSO is less clear. Neither the amplitudes nor descent rates of the QBO are significantly different between EN and LN phases, according to a Welch t-test. Interestingly, the only significant difference in the model

is that descending westerlies are slower in Neutral ENSO months compared to EN or LN conditions, perhaps suggesting that characteristics of tropical wave activity may be different in ENSO phases compared with neutral years (e.g. Geller et al., 2016). The model results therefore suggest that there is little influence of ENSO on the descent rate and amplitude of the QBO in the GC3 N216-pi simulation; this result was also found for the lower resolution simulations (not shown). In addition, no evidence was found that strong warm ENSO events change the phase of the QBO in this model, in contrast to the findings of

Christiansen et al. (2016). This finding of a null influence of ENSO on the QBO agrees well with the results of Serva et al. (2020) that examined these relationships in an older version of the HadGEM model.

The reversed possibility, i.e., that the QBO may somehow influence ENSO is now examined, first, by estimating whether the frequency of ENSO events significantly depends on the QBO phase. Table 1 documents the frequency of ENSO events in each QBO phase from observations/reanalysis and the three model simulations. A higher frequency of EN events during QBOW

and of LN during QBOE has been observed from 1979 to 2020 (Taguchi, 2010; Liess and Geller, 2012), but the opposite is

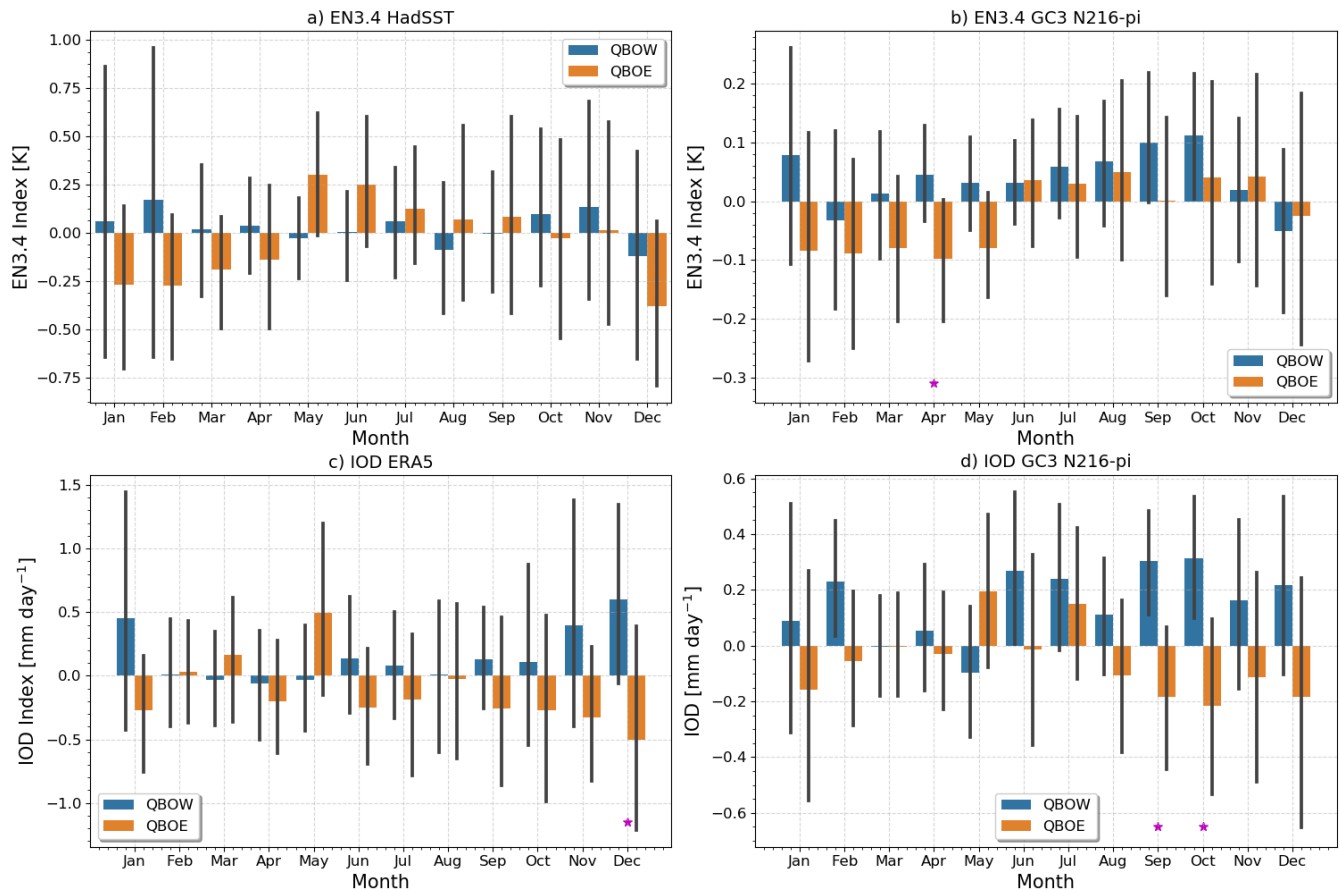

**Figure 7.** Monthly-mean (a-b) EN3.4 and (c-d) IOD indices separated per QBO phase in (a, c) observations/reanalysis ERA5 and HadSST (1979-2021) and (b, d) GC3 N216-pi. The error bars show the standard deviation of each index for each month and significant differences between QBO W and E months are highlighted with a ∗ at the bottom of each panel.

diagnosed if the period is extended to 1953-2020, in agreement with previous sections and studies (Domeisen et al., 2019). Probability density functions (PDFs) were constructed for the model data using 36-yr samples with replacement. In addition to a Welch's t-test, a Kolmogorov–Smirnov (KS) test was used to evaluate if the PDFs of an event frequency (e.g. EN) were significantly different for each phase of the QBO.

The results show significant differences, according to both KS and Welch tests, for each ENSO phase in the three model simulations. EN events are more frequent under QBOW conditions than under QBOE in both observations and model datasets. LN events are equally frequent in both QBO phases in the HadSST dataset but in GC3 N216-pi they are more frequent under QBOE than under QBOW. Note that the frequencies of LN and EN under neutral QBO conditions in the model were ±0.26 month month$^{-1}$ in both cases.

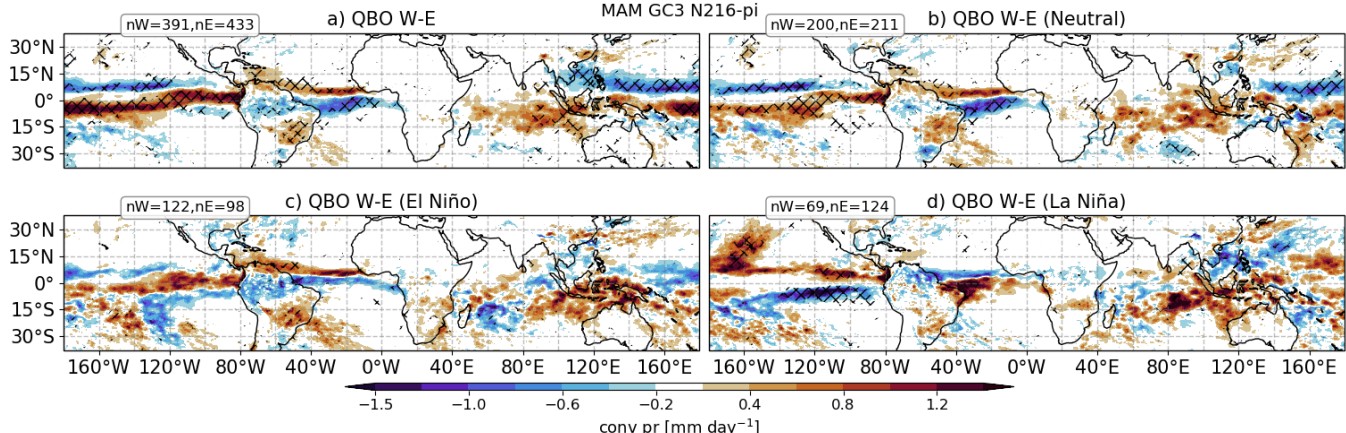

**Figure 8.** Composite convective precipitation differences (QBO W-E) in MAM for GC3 N216-pi (a) all the months and (b) Neutral ENSO conditions only, (c) El Niño and (d) La Niña. The sample size of each composite is noted in the top left corner of each panel. Statistically significant differences to the 95% confidence level are shown through the hatching.

Figure 7a,b shows the EN3.4 index amplitude and interannual standard deviation as a function of each month from the HadSST dataset and the GC3 N216-pi simulation, separated for each phase of the QBO. From this we can examine, for example, whether any QBOW minus QBOE differences in ENSO characteristics arise primarily from one QBO phase or the other (i.e. a non-linear response) or whether both phases contribute equally to the response difference.

In the observed period of 1979-2021, the EN3.4 SST is negative from December to April under QBOE. These results are consistent with the analysis of ENSO frequency in Table 1, which shows more frequent La Niña events under QBOE and El Niño events under QBOW. In the model, the mean EN3.4 index is frequently positive under QBOW and also negative from December to April under QBOE. In both cases, the strength and sign of the differences varies seasonally, for example, the only month where the EN3.4 index is statistically different is April.

The previous sections have demonstrated that the mean QBO response is affected by an uneven frequency of ENSO events in each QBO phase. In addition, evidence was found of non-linear ENSO impacts associated with the QBO or in other words, that the QBO response can also be a function of ENSO. This non-linearity can be observed in Figure 8 where the QBO composite differences in convective precipitation in MAM are shown using all years, Neutral ENSO years and EN or LN years. While the broad nature of the QBO signal remains similar, the details differ depending on the phase of ENSO (8c,d). For example, the Atlantic and Pacific ITCZ responses are opposite during LN compared to Neutral and EN years. The total equatorial Atlantic response is then a result of the combination of EN and Neutral years which is dampened or obscured by the LN years.

Another prominent feature of the composite of all years is the off-equatorial West Pacific positive response which is only observed for Neutral years. This result suggests that some ENSO impacts, e.g., over the Atlantic basin, are different depending on the QBO phase. In GC3 N216-pi, this effect is more readily observed during MAM but similar results are found for the

other two models in DJF (Figs. S6 and S7). The implication of these results would be that, in some cases, regression analysis may not be the right approach because the QBO surface impacts may be non-linear.

In the previous sections, the precipitation and SST analyses also show suggestive evidence of a relationship between the QBO and the Indian Ocean, in both the observations and the model. The link between the QBO and the IOD event frequency have been analysed in the same way as for the ENSO index and a significant relationship is confirmed (Table 1 and Figs. 7c-d). The IOD event frequency is also markedly different depending on the QBO phase, with positive events more frequently observed in the westerly phase of the QBO and negative events found more frequently under QBOE, both for ERA5 and the model simulations (Table 1). The monthly-mean values in Fig.7c-d for the model indicate a more frequently positive IOD index under QBOW and a negative index for QBOE, and these differences are statistically significant in September and October. The GC3 N96-pi and UKESM N96-pi results are very similar (Fig. S5) and the differences are also significant in SON.

This section demonstrates statistically robust links between the IOD and ENSO, and the QBO. These phenomena (ENSO and the IOD) are intertwined by pan-tropical teleconnections through zonal circulations (Cai et al., 2019), and interact with monsoons and the ITCZ. For that reason, the following section explores more closely the links between the QBO and features of the circulation in the tropics.

## 3.4 ITCZ, monsoons and the tropical overturning circulation

This section investigates the QBO impacts on the ITCZ, monsoons and the Walker circulation. The previous sections demonstrated a robust link in the simulations between ENSO and the QBO, and for that reason, this section presents results when the influence of ENSO has been removed by using ENSO Neutral composites (NN). Model biases in the representation of the migration and dynamics of the ITCZ, measured by zonally averaged convective precipitation in the Pacific and Atlantic sectors (Fig. 9a-b), are highly relevant since these biases may modify any physical mechanisms of the QBO over convection. These biases can be characterized by a southward shift of the simulated Atlantic ITCZ in DJF and MAM and a wider extent of the Central Pacific ITCZ compared to ERA5 leading to a "double-ITCZ" in the Pacific during boreal winter (García-Franco et al., 2020). Note that the magnitude of the biases is almost as large as the climatological values during boreal winter.

The monthly-mean QBO W-E zonal-mean convective precipitation differences in the Pacific and Atlantic ITCZ regions during NN ENSO conditions (Figure 9) show that the ITCZ impacts are seasonally dependent. While there are no clear differences in the Atlantic sector for ERA5 in any month, in GC3 N216-pi there is a significant northward shift of the ITCZ from April to June, which is likely associated with the warm SST anomalies found in this season in the northern tropical Atlantic (Fig. 3).

The differences in the Pacific sector for ERA5 show a very noisy and mixed result. However, in GC3 N216-pi, a southward shift of the ITCZ is observed from February to July, maximized in the MAM season. Very similar results for the Atlantic and Pacific sectors were observed for the other two simulations (Fig. S8). Note that these results are for ENSO Neutral conditions, and for all years the link between QBO and ENSO is evident (Fig. S9).

Observational (Collimore et al., 2003; Liess and Geller, 2012; Gray et al., 2018) and modelling (Giorgetta et al., 1999; Garfinkel and Hartmann, 2011a) evidence has documented links between the QBO and monsoon regions. However, the results in the previous sections (Fig. 1) show little-to-no robust effects of the QBO on precipitation over land in the simulations. The

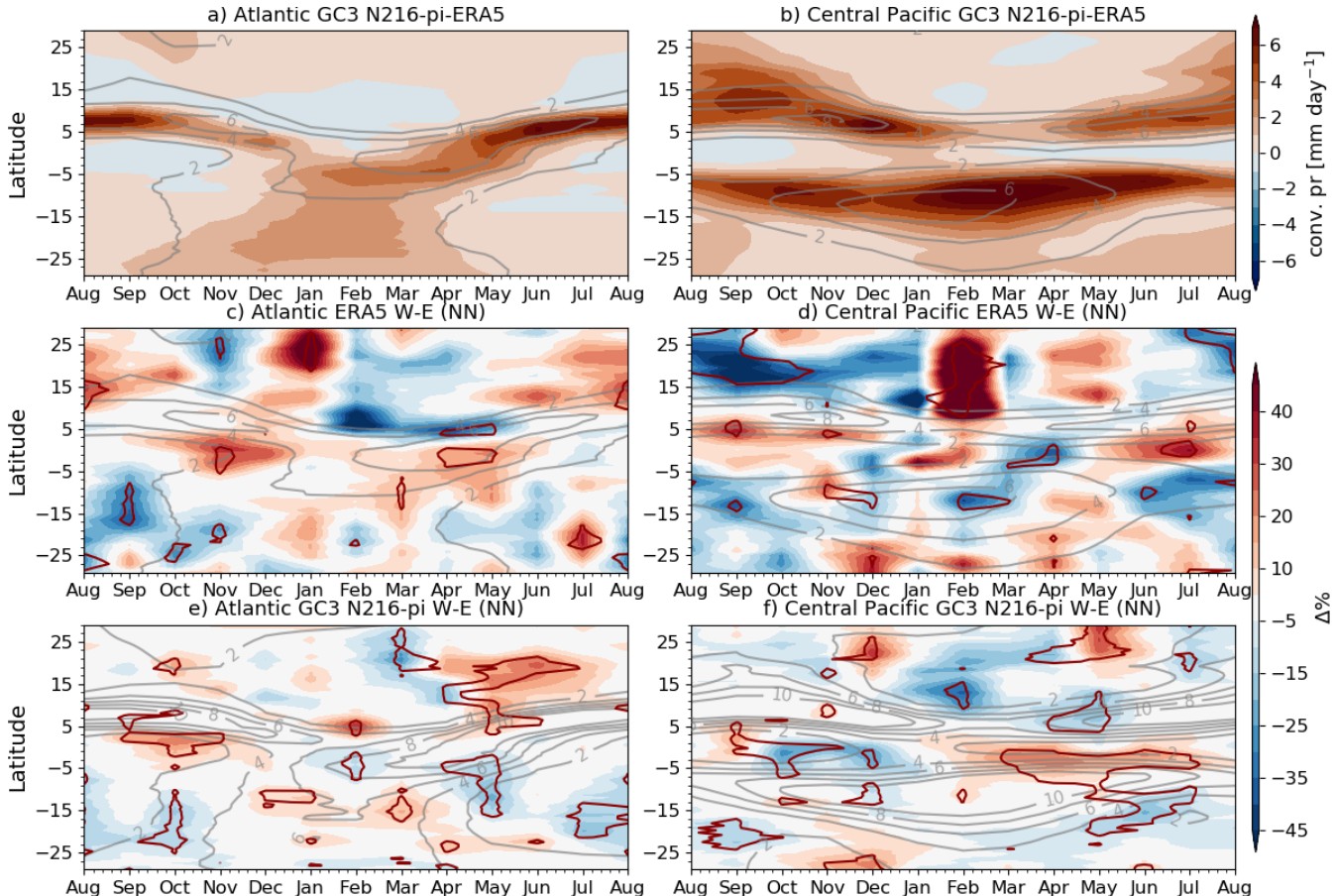

**Figure 9.** (a, b) Zonal mean biases in convective precipitation in GC3 N216-pi compared to ERA5 in the (a) Atlantic [60°W-20°W] and (b) Central Pacific [180°W-140°W] sectors. These sectors that represent the Pacific and Atlantic ITCZ were chosen based on the regions that exhibit a signicant impact in Figure 5. (c-f) Monthly and zonal mean QBO W-E percent (%) differences during NN ENSO conditions in convective precipitation where the absolute difference is weighted by the climatological value at each latitude and month. The line-contour (red) depict differences that are statistically significant to the 95% level according to a bootstrapping test and the grey lines show the climatological values.

precipitation response over land is examined more closely by analysing regions that fit the concept of the global monsoon. For this purpose, a monsoon region is defined as a region in which over 55% of the total annual rainfall is observed or simulated in the respective summer season and the summer-winter precipitation difference is higher than 2 mm day$^{-1}$ (Wang and Ding, 2008; Wang et al., 2017, 2021). After defining these regions, the QBO W-E differences, during ENSO Neutral-only, are computed for JJAS and DJFM for Northern and Southern Hemisphere monsoons, respectively for GPCC (1953-2020) and GC3 N216-pi).

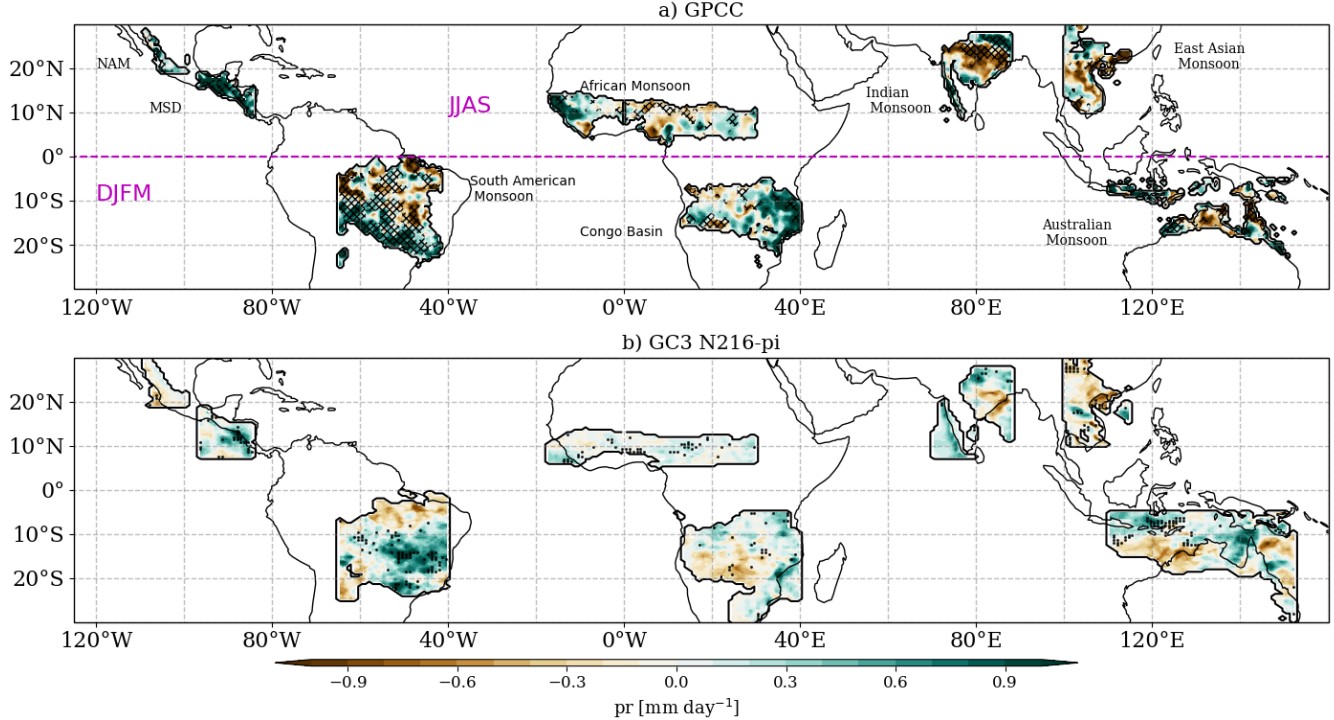

**Figure 10.** Total precipitation differences in monsoon regions between QBO W-E phases under Neutral ENSO conditions-only for a) GPCC (1953-2020) using the FUB QBO index and b) GC3 N216-pi. For monsoon regions in the Northern hemisphere, differences are shown for the JJAS period, whereas for Southern Hemisphere monsoons, results are shown for DJFM. Hatching and dots indicate differences that are statistically significant to the 95% level. NAM and MSD stand for the North American Monsoon and the Midsummer Drought (García-Franco et al., 2020), respectively.

Figure 10 shows that precipitation response over monsoon regions is relatively weak in GPCC and GC3 N216-pi. In the South American and Indian monsoon regions, for example, both positive and negative significant differences are observed indicating no region-wide coherent impact. This lack of spatial coherency suggests that regional dynamics are likely important. In GC3 N216-pi, in the South American monsoon region, the QBO W-E differences indicate a significantly wetter region in South America, where the South Atlantic Convergence Zone is located (Carvalho et al., 2004; Jorgetti et al., 2014). Similarly, wetter conditions over southern Mexico and Central America (the Midsummer Drought region) (García-Franco et al., 2020) are observed during QBOW compared to QBOE in GPCC and, albeit much weaker, in GC3 N216-pi.

The climatological biases in the representation of the monsoon dynamics by this and other climate models (e.g. in South America García-Franco et al., 2020; Coelho et al., 2022) could mean that the impacts seen in Figure 10 are model-dependent and our analysis of the lower resolution simulations (Fig. S10) indicates that some of these impacts are also resolution-dependent. This reinforces the notion that the mean representation of the dynamical features of each monsoon by a model configuration is important for any subsequent response to the QBO. Nevertheless, this analysis shows that in the model the

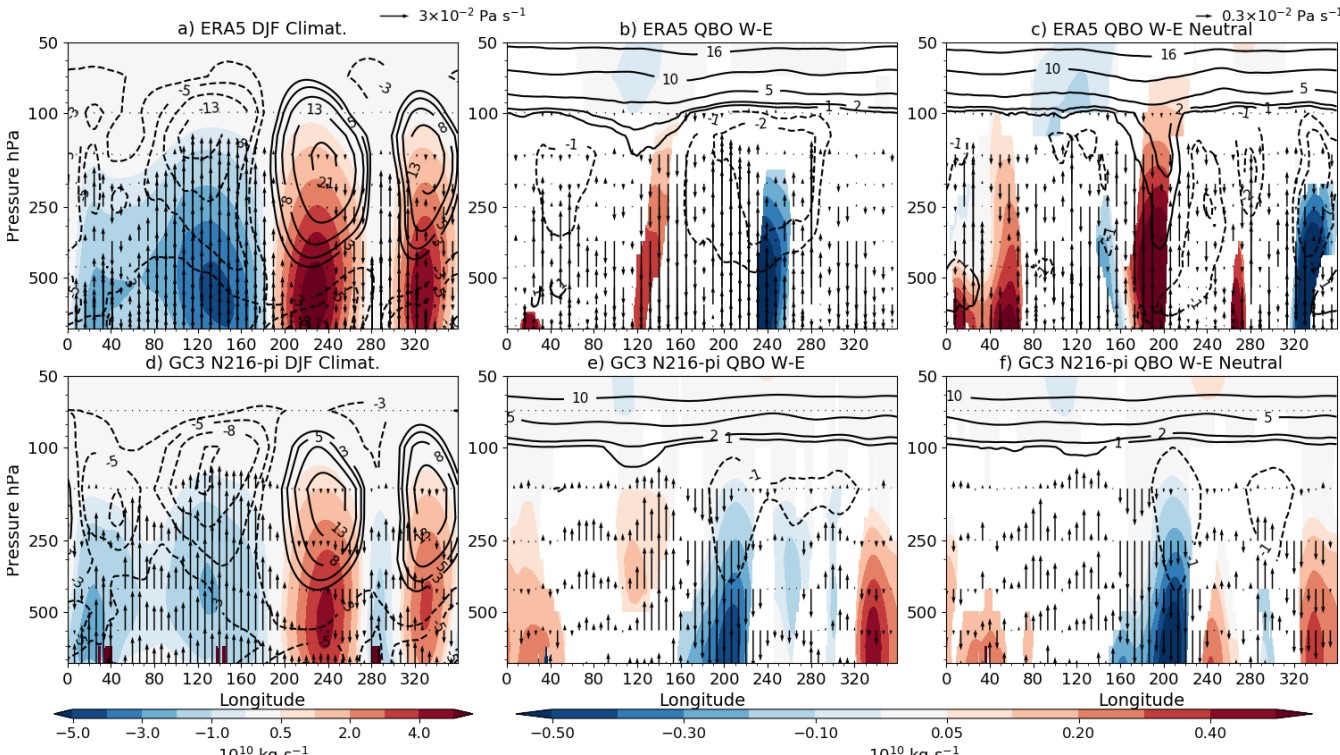

**Figure 11.** (a, d) Climatological mean-state of the Walker circulation, depicted through the zonal streamfunction ($\psi$) in shading, the zonal wind (contours), and vertical velocity ($\omega$ [Pa s$^{-1}$], vectors) during the DJF season in a) ERA5 and (b) N216-pi. (b, c, e, f) show W-E composite differences, during DJF, for the same variables and only statistically significant differences (95% confidence level) are shown. (c, f) are as in (b, e) but considering Neutral ENSO periods only. Example vector sizes for $\omega$ are given in the top right corners of a and c. Note that the colorbar and vector sizes are different for the climatology plots (a,d) than for the anomaly plots.

QBO impacts on land convection are weaker than on oceanic convection, suggesting that SST feedbacks may be important for the QBO response in the model.

A number of studies have suggested a link between the QBO and the Walker circulation to explain the zonally asymmetric nature of the QBO surface impacts in the tropics (e.g. Collimore et al., 2003; Liess and Geller, 2012). Therefore, the relationship
375  between the Walker circulation and the QBO is examined to evaluate this hypothesis through the use of the zonal streamfunction (Yu and Zwiers, 2010; Bayr et al., 2014; Eresanya and Guan, 2021), defined as:

$$\psi = 2\pi \frac{a}{g} \int_{0}^{p} u_D \, dp, \tag{1}$$

where $\psi$ is the zonal streamfunction, $u_D$ is the divergence part of the zonal wind, $a$ is the Earth's radius, $p$ is the pressure coordinate and $g$ the gravitational constant. The divergent component of the zonal wind ($u_D$) is computed by solving the

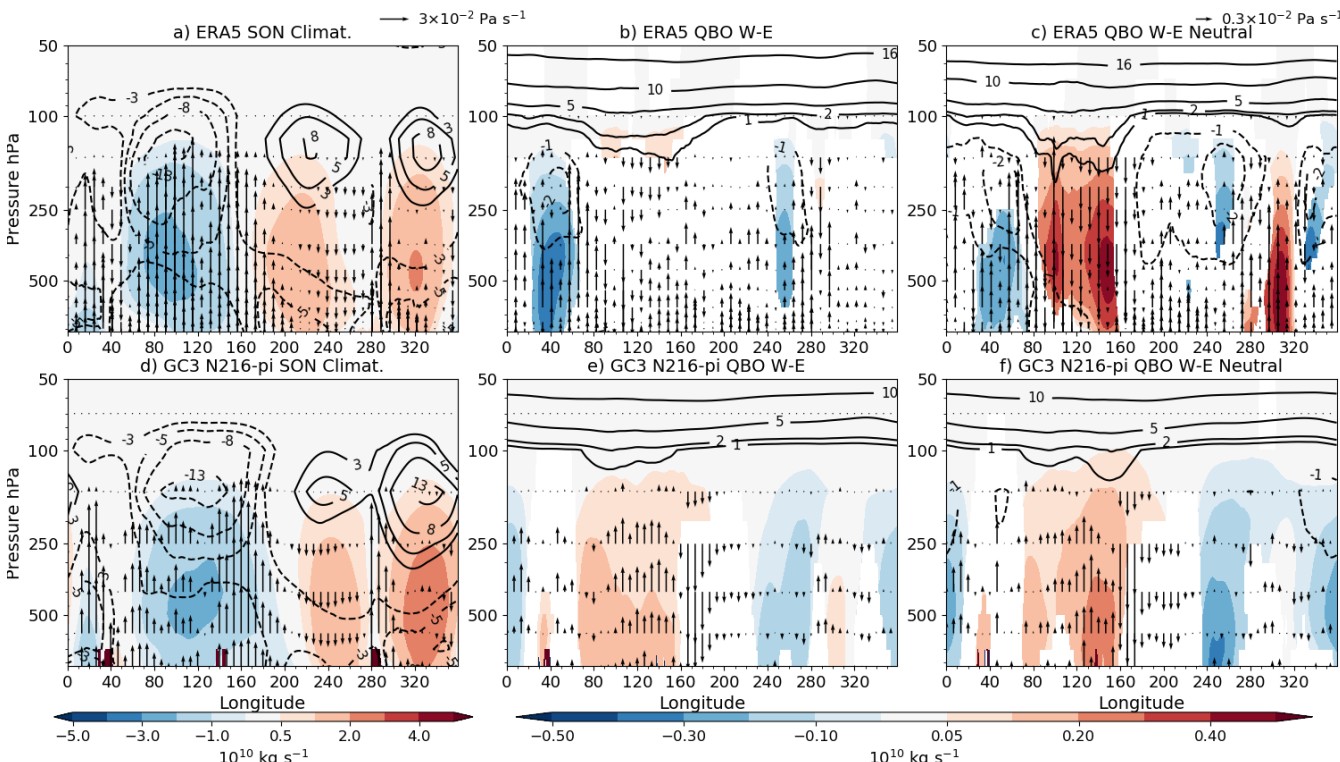

**Figure 12.** As in Figure 11 but for the SON season.

Poisson equation (Eresanya and Guan, 2021) for the velocity potential using the python package windspharm (Dawson, 2016) that employs spherical geometry. The streamfunction is calculated by first averaging over the equatorial band of 10°S-10°N and integrating from the top level of each dataset to the surface.

QBOW minus QBOE composite differences in DJF show that the streamfunction in the eastern Pacific [220-260°E] is significantly weaker during QBOE than during QBOW in ERA5 and GC3 N216-pi (Fig. 11). In the model, these streamfunction differences are significant even low in the troposphere. The zonal wind at upper-levels (300-100 hPa) is also weaker in QBOW compared to QBOE at 200°E in both model and reanalysis. In GC3 N216-pi, the negative $\psi$ difference is accompanied by descending motion anomalies in the 170-220°E region, whereas anomalous ascent is observed in the Maritime continent and Indian Ocean. The differences in the other simulations agree with the results of GC3 N216-pi (not shown).

In boreal fall (Fig. 12), the differences are also significant and can be linked to the relationships found between the IOD and the QBO. Specifically, significant positive differences (W-E) in the streamfunction are found in the eastern Indian Ocean and maritime continent and negative differences in the eastern Pacific. In GC3 N216-pi, vertical velocity anomalies indicate stronger ascent in the western Indian Ocean and in the Maritime continent whereas weaker ascent anomalies are found in the eastern Indian Ocean. These results agree with positive IOD indices found in QBOW and a mean negative index during QBOE.

The rightmost panels in Figures 11 and 12, in which only Neutral ENSO months are considered, suggest that this relationship between the QBO and the Walker circulation occurs regardless of ENSO events for GC3 N216-pi. However, some ENSO-QBO superposition can be seen from the plots, e.g., in ERA5, removing ENSO events changes the sign of the response. This effect is likely due to the small sample size in the observational record when only neutral months are considered. The weakening of the Walker circulation under QBOW compared to QBOE is also seen in the other model configurations (Figs. S11 and S12). These results highlight links between the large-scale overturning circulation and local responses which may explain the zonally asymmetric results found in previous studies and in early sections of this paper.

## 4   Summary and discussion

Analyses of observational records of clouds and precipitation in the tropics suggest links between the stratospheric QBO and tropical deep convection (Collimore et al., 2003; Liess and Geller, 2012; Gray et al., 2018). However, the short observational record available and the confounding influence of ENSO and its teleconnections limits the robustness of any analysis seeking to explore these links and possible mechanisms of interaction between the QBO and the tropical troposphere. This study investigates the tropical signature of the QBO in the 500-year-long pre-industrial control CMIP6 experiments of the Met Office Hadley Centre Unified Model, with a focus on the HadGEM3 GC3.1 N216 simulation.

Composite and regression analyses were used to demonstrate the presence of a statistically significant link between the QBO and precipitation in the tropics. These impacts were found in the position and strength of the Pacific and Atlantic ITCZ, as well as in the Caribbean Sea and the Indian Ocean. The QBO signal was found to be zonally asymmetric, with the more robust and largest differences over the oceans, suggesting the possibility of SST feedback processes and a role for the Walker circulation. Impacts over monsoon land regions were found to be much weaker.

A similarity between the QBO precipitation response pattern and positive ENSO events raised the possibility of an aliasing of ENSO and QBO signals in the model and observations. The interaction of the QBO and ENSO signals was extensively explored. When only ENSO-neutral years are analysed the QBO signal remains essentially unchanged, ruling out the possibility of a straightforward aliasing of ENSO events with the QBO phase selection, a result that was confirmed by multivariate regression analysis (Fig. 5). Additionally, this study examined the possibility that the QBO response in regions dominated by ENSO teleconnections could be due to an influence of ENSO on the QBO, rather than a real downward impact of the QBO itself. This upward interaction could be via modulation of tropical wave generation, as has been proposed previously (Schirber, 2015). The model was shown to succesfully simulate the well-known differences in QBO descent rates in which the QBOW phase descends more rapidly than the QBOE phase. However, there was no evidence in the model for an ENSO influence on the rate of descent or amplitude of either QBO phase.

While recognising that linear diagnostics are unable to provide specific evidence of cause and effects, our analysis demonstrated that the ENSO-QBO relationship is statistically significant in the model. The frequency of ENSO events in each phase of the QBO was first explored. In observations, the ENSO-QBO relationship shows decadal variability; in recent decades El Niño events have been found to occur more frequently in QBOW years and La Niña events are more frequently found in

QBOE years (Taguchi, 2010). However, the use of radiosondes and a reconstruction by B07 highlighted that this relationship has changed sign over different periods from 1930-2020.

In the model, more frequent El Niño events are found under QBOW and La Niña events under QBOE. However, this relationship is also non-stationary and in some 30-50 yr periods, the opposite QBO-ENSO relationship can be found. The examination of the month-by-month EN3.4 amplitude in the model showed that the interaction between QBO and ENSO is far from linear, since the amplitude dependence on QBO phase was asymmetric and strongly seasonally dependent. The non-linearity of the QBO-ENSO interaction was confirmed using composite analyses that showed different QBO signal patterns during El Niño years as compared with La Niña years.

In addition to the QBO-ENSO link, the model analysis also highlighted a statistically significant QBO signal in precipitation over the Indian Ocean, raising the possibility of an interaction with the IOD. In boreal fall, the IOD index, a measure of the zonal gradient of convective precipitation in the Indian Ocean, was found to be anomalously positive in QBOW years and anomalously negative in QBOE years, both for ERA5 and the three model simulations.

Finally, previous studies have proposed that the QBO may influence the mean-state of the Walker circulation, which could explain the zonally asymmetric nature of the QBO signal in precipitation in the tropics (Collimore et al., 2003; Liess and Geller, 2012; Hitchman et al., 2021). The modelled Walker circulation was found to vary by up to 10% between QBO phases, even when the effect of ENSO events was taken into account. Specifically, the Walker circulation was found to be weaker during QBOW than during QBOE.

Most of the results in this study agree with previous analyses of models and observations (Gray et al., 2018; Hitchman et al., 2021; Serva et al., 2022). However, Rao et al. (2020) found very different responses in a set of CMIP5/6 models, including HadGEM3 and UKESM1. These differences can be explained, first, by the fact that the study of Rao et al. (2020) examined $\approx$150-yr long simulations with historical forcings, in contrast to the 500-yr control simulation (no external forcing) examined by this study, and, second, because Rao et al. (2020) used a different QBO index based on the 30 hPa winds compared to this study (70 hPa). The 30 hPa level index captures very little of the QBO-driven temperature variability near the tropopause (Fig. S13) compared to the 70 hPa index, and the use of only one ensemble member to diagnose the precipitation response in a simulation with time-varying external forcing can result in different (and misleading) impacts compared to the ensemble-mean of all available members (Fig. S14).

The role of model biases for these results must be emphasized and the results treated with caution, since a different representation of the stratosphere or the troposphere may control the extent and location of the QBO influence. Tropospheric biases, e.g., in the strength or position of the ITCZ (Fig. 9), and stratospheric biases such as the weak amplitude of the QBO in the lower stratosphere (Bushell et al., 2020), may limit the robustness of these results and may mean that the impacts diagnosed in this study may be different in another model with different biases.

The exact nature of the relationship between the QBO and tropical deep convection remains to be well understood. The robust connections diagnosed in this study warrant further analysis as the causality of these connections could not be addressed fully here. Targeted model experiments citep[see e.g.,][]garfinkel2011,martin2021 would help to investigate hypotheses about

causal mechanisms, such as the static stability mechanism (Hitchman et al., 2021), in order to disentangle the direction of causality between the tropical stratosphere and troposphere.

*Data availability.* ERA5 reanalysis data are available from the Copernicus Climate Change Service Climate Data Store (Hersbach et al., 2018) at https://doi.org/10.24381/cds.adbb2d47 and https://doi.org/10.24381/cds.bd0915c6. The FUB dataset was obtained from https://www.geo.fu-berlin.de/met/ag/strat/produkte/qbo/qbo.dat whereas the reconstruction can be obtained at https://climexp.knmi.nl/getindices.cgi?WMO=BernData/qbo_90&STATION=QBO_90&TYPE=i&id=someone@somewhere. The HadSST 4.0 dataset is available at https://www.metoffice.gov.uk/hadobs/hadsst4/data/download.html. The GPCP v2.3 was downloaded from https://psl.noaa.gov/data/gridded/data.gpcp.html. CMIP6 simulations used in this study are available from the Earth System Grid Federation of the Centre for Environmental Data Analysis (ESGF-CEDA; https://esgf-index1.ceda.ac.uk/projects/cmip6-ceda/, WRP, 2019, last access: 22 Oct 2021).

## Appendix A: Methods

### A1 Regression analysis

The simple linear regression model can be written as:

$$Y(t) = X_0 + X_i(t)\beta_i + \epsilon, \tag{A1}$$

where $Y$ is the measured or dependent variable, $X_0$ is a constant coefficient, $\beta$ is the regression coefficient between $X$ and $Y$ and $\epsilon$ represents random error or a residual. In all cases, the models were solved using an ordinary least-squares (OLS) method. A multivariate regression model was used to study the joint effect of two or more predictors, in this case ENSO and QBO indices, over a variable $(Y)$, in this case precipitation, such that the model can be written as:

$$Y(t) = X_0 + \sum_{j=1}^{N} X_j(t)\beta_j + \epsilon \tag{A2}$$

where $X_j(t)$ is any predictor with an associated regression coefficient $\beta_j$.

As in previous studies (Gray et al., 2018; Misios et al., 2019), the regression coefficient can be rescaled to evaluate the total effect that a predictor $(X_j)$ can have on the variance of the measured variable $(Y)$ using the standard deviation $(\sigma_j)$ and the maximum $(X_{j,max})$ and minimum $(X_{j,min})$ values of $X_j$ so that the rescaled coefficient $\beta'_j$ can be written as:

$$\beta'_j = \beta_j \frac{X_{j,max} - X_{j,min}}{\sigma_j}. \tag{A3}$$

*Author contributions.* JLGF, SO and LJG designed the scope and model analysis. JLGF conducted the analyses. RC and ZKM contributed in the interpretation of results. All authors were fully involved in the revisions and the preparation of the paper.

*Competing interests.* The authors declare no competing interests.

*Acknowledgements.* J.L.G.F was supported by an Oxford-Richards graduate scholarship under Wadham College, and by a Met Office-Partnership PhDCASE studentship. Z.K.M acknowledges support from the National Science Foundation under Award No. 2020305. LG and SO wish to acknowledge support from the United Kingdom Natural Environment Research Council and National Centre for Atmospheric Science.

490

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
