# Peer review of "The tropical route of QBO teleconnections in a climate model"

_Weather and Climate Dynamics, 2022_

## Author Comment (AC1)

**Manuscript # WCD-2022-14 Response to Editor and Reviewers**

García-Franco, JL., Gray LJ., Osprey S., R. Chadwick and Z. Martin

Atmospheric, Oceanic and Planetary Physics, Department of Physics, University of Oxford. Parks Road Oxford, United Kingdom OX1 3PU email: jorge.garcia-franco@physics.ox.ac.uk

**1. Reviewer 1**

This work by Garcia-Franco et al. looks at the relationships between the QBO and tropical climate in observations and centennial pre-industrial CMIP6 simulations with one coupled climate model. The connections are difficult to diagnose from observations so long simulations are useful. The paper is overall interesting and covers many topics, but the authors should check the consistency of the symbols and names used (see comments by line).

Many thanks for your time reviewing our manuscript and for your constructive criticism. We have addressed your comments and we hope you find the manuscript to be improved and suitable for publication. In this document we provide a point by point reply to the comments of both reviewers, where your comments are shown in black, and our response in blue.

**1.1. Major comments**

1.1.1. It can be confusing to read different acronyms for the same quantities. The units reported in the plots should be verified.

The revised manuscript now consistently uses acronyms defined in the method section for the easterly (E) and westerly (W) phases of the QBO, as well as El Niño (EN), La Niña (LN) and Neutral (NN) ENSO conditions. The units in all plots now consistently use [], as suggested below by this reviewer.

1.1.2. Given the central role of model simulations, more information on its skill at simulating QBO and ENSO should be provided. For example, how realistic is the QBO amplitude at 70 hPa for this specific model?

The first version of the manuscript discussed the performance of the model in the tropical troposphere based on available CMIP6-wide assessments in section 2.2. The revised manuscript adds a second paragraph (at the end of sectxtended ENSO seasonion 2.2) that provides specific information about the model representation of ENSO and the QBO. Weaknesses and strengths of the model are mentioned in this section. For example, the revised manuscript also discusses the realism of the QBO amplitude with emphasis on bias at the 70 hPa level Bushell et al. (2020). This bias is key for our study and is discussed further in the final section of the manuscript. 1.1.3. Apart from composite differences, some climatologies should be discussed.

The relevance of the climatological representation of the tropical features is stressed more in section 3.4 as well as in the discussion section in the revised manuscript.

- 1.1.4. In the introduction reference to Geller et al., 2016 on gravity wave changes would fit.We have added the reference mentioned in several instances of the revised manuscript.
- 1.1.5. Model-dependence of the results should be stressed, since different configurations of a single model are analysed and QBO/SST biases may play a big role. The causality analysis on how the QBO influences ENSO is not very convincing as it stands.

The discussion section now emphasizes that these results could be model and bias-dependent. We agree that this manuscript does not provide a causality analysis on the QBO-ENSO relationship. However, given the non-stationarity of the QBO-ENSO relationship in the observations and the lack of any model study reporting that this relationship exists in a climate model, our findings are relevant and strongly emphasize the need for targeted model experiments that will causally disentangle these relationships.

1.1.6. I guess the authors should also say something about the frequency of LN/EN events during neutral QBO (QBO-N).

The revised manuscript now makes a note of the frequency of LN/EN events during neutral QBO periods, which are 0.258 and 0.263 month month-1, respectively, so we report them to be roughly evenly distributed.

1.1.7. The section about monsoons should be revised and maybe shortened, since QBO surface impacts may be very dependent of any QBO bias. For example, Giorgetta et al. 1999 (cited) nudged to QBO, so it was realistic in their case.

The section on monsoons and the ITCZ has been revised to emphasize the importance of model biases, both in the troposphere and the stratosphere. The section on monsoons has been shortened to two paragraphs. A paragraph in the discussion emphasizes how these impacts could be model and bias-dependant.

1.2. Specific comments

L52, maybe 'on the convective process'? Reworded as suggested.

L55, define 'CMIP', rephrasing L62

**Done**

L63, are GWs tied somehow to sources?

The model can sustain GWs of sufficiently large scale to be resolved by the resolution of the model, these resolved waves are tied to convective sources and the mean flow (Walters et al., 2019), however, waves of scales that are too small for the model grid are parametrised using a spectral sub-grid parametrisation scheme (Scaife et al., 2002) which in the current version of the model is parametrised so that the wave momentum flux is a function of total precipitation at each grid-point (Bushell et al., 2015).

L76, both monthly means?

Yes, we have specified this in the text.

L82, is there a reason for not using the standard  $0.25 \times 0.25$ ?

We have chosen to use a coarser resolution than the default  $0.25 \times 0.25$  to balance the computational cost of the vertical integrals using daily means for the mass overturning circulation and maintaining a reasonable resolution ( $0.75 \times 0.75$ ) for the rest of diagnostics.

L83, it is a bit strange to put the (generic) link only for ERA5; I would move to data availability with direct links for all datasets (for ERA5 https://cds.climate.copernicus.eu/cdsapp#!/dataset/ reanalysis-era5-pressurelevels-monthly-means?tab=overview) and proper citation https://confluence. ecmwf.int/pages/viewpage.action?pageId=197704114

We have moved the link to the data availability section, specifying the links (through the DOI) to monthly and hourly download landing pages.

L90seq, define 'N' and 'ORCA' for the components resolution.

We have defined 'N', whereas ORCA is a standard and generic name for the tripolar mesh grid used in the NEMO model.

L91, UKESM or UKESM1?

We have chosen to use UKESM1 as the model name and reference the pre-industrial control experiment with the acronym UKESM N96-pi throughout the study.

L94, So 3 simulations in total? Would be good to state that you have two models with lower resolution and one with better resolution, which is the main interest.

We have made this point clearer in the text.

L96, I would move to 'data availability' or similar

Done.

L98, Here or later I would add something about some relevant model properties (e.g. both models have more spectral power in 2-3 years compared to observations). Also ocean resolution seems to be important for mean biases, and the realism of the ITCZ should be mentioned as well.

In the last paragraph of section 2.2 we describe the strengths and weaknesses of the Met Office models, with particular emphasis on the medium-resolution configuration of HadGEM3 GC3.1 The revised manuscript includes details on the biases of the QBO (e.g. more power at longer periods than observations) and on the realism of the ITCZ placing these biases in the context of current state-of-the-art CMIP6 models. The oceanic resolution of the model is stated in the first paragraphs of section 2.2; the relatively high oceanic resolution in GC3 N216-pi may be part of the reason why the GC3 N216-pi is top ranked in most model assessments in the tropics. However, we cannot comment further as we are not aware of a study that systemically analyses how oceanic resolution impacts model's performance in the tropics in CMIP6.

L105, What about UKESM1? Not sure why only HadGEM is mentioned.

There are several reasons for this: (1) the results of the manuscript are primarily presented for the medium resolution configuration of HadGEM3. (2) The assessment studies cited in this paragraph only seldom used the UKESM model so we cannot place UKESM into the wider context of tropical biases as we did for HadGEM3. (3) UKESM uses the same dynamical core as HadGEM3 which means that in many contexts of tropical climate, the climatological biases are very similar to HadGEM3 in pre-industrial control experiments (García-Franco et al., 2020).

L111, years or months?

Months. Corrected

L117, Which levels? Above you just mention 70 hPa.

We have specified that we use the range of 10-70 hPa.

L119, '1' and '2' are subscripts

Done.

L124, The product you use (GPCP?) for this index is providing convective and stratiform precipitation separately? Or is it a total precipitation? If not, remove convective (here and also in all instances following). Can you explain why using a precip-based IOD index rather than the standard SST-based one? Please add a reference if it was used before.

We use the convective precipitation from ERA5, not from GPCP, to better diagnose the effect of the QBO on deep convection. This index was used because the dipole signal is diagnosed for precipitation and not for SST in the model data.

L125, I'd use same style for EN3.4, with []

Done.

L133seq, This symmetry seems strange (given the ENSO asymmetry and QBO stalling) can you provide numbers?

Good point, indeed neither ENSO or QBO numbers are symmetrical. The point of these sentences was to highlight how the length of these simulations renders large composite sizes and provided rough numbers. The revised manuscript states the specific numbers for each phase to clarify this point.

L135, This is 'observed' for ERA5? Can you provide the values for HadSST? It is useful to compare model/observation statistics.

SSTs and ENSO indices are obtained from HadSST, so these 'observed' quantities are obtained using SSTs from HadSST and QBO winds from ERA5.

L140, Maybe start with 'When estimating correlations, they are...'

Done.

Fig 1, 'mm day-1' in brackets, or move 'pr' to title

Done.

L159, Please comment on the ITCZ realism.

This is a good point, ITCZ biases in the ITCZ limit the realism of the diagnosed impacts in the model, the reader is referred the reader to section 3.4 which describes and discusses the biases and their influence on the diagnosed response.

L162, Add reference

Done.

L206, I guess would be useful to have a table in the method section with the different numbers for ENSO and QBO. Why 120, does it have a special meaning?

We have removed the 120 count reference.

L209, But the wet anomaly in the Pacific and dry in the Atlantic are more marked with ENSO included. This is also seen in Fig5.

Good point. The revised version now makes this distinction, i.e., there are some regions that exhibit a robust QBO response that is independent from ENSO, whereas the response in the equatorial Pacific and Atlantic is a function of both ENSO and the QBO.

Fig5, If regression coefficients are re-scaled (caption), then a prime is missing in a&d titles. See Supplement as well.

Done.

L214, (1) - > (Fig. 1)

**Done.**

L216, it was EN3.4 before The revised manuscript now uses EN3.4 as the acronym for the EN3.4 index throughout.

L221, why no significance in FigS3?

The significance hatching has been added to this plot.

L225, mention Gray et al., 1992

Done.

Fig6, I'd use E and W for QBO in (b). Moreover I would define once all the acronyms (EN, LN, E, W) in the methods and be consistent throughout (no 'ea', EN3.4, etc.). Suggest NE or NN for Neutral ENSO. Moreover, would it be easier to read the plot ordering the boxplots as LN/NE/LN? Why not showing E and W phases separately for the amplitude?

The revised figure uses E and W as the rest of the manuscript (no 'ea'). The rest of the manuscript now uses the acronyms suggested by the reviewer consistently.

Unfortunately, we didn't understand the suggestion "would it be easier to read the plot ordering the boxplots as LN/NE/LN", perhaps the reviewer meant to say the NN composite should be in the middle of LN and EN boxplots. We have created such a figure and found little improvement to the readability of the figure.

Finally, the amplitude was not separated in the figure because there was no ENSO modulation of the amplitude regardless of QBO phase, so we chose to use the version of the figure that is more concise.

L238, Have you stated which level are descent rates for? From the methods I got that the amplitude is integrated in the 10-70 layer, but descent rate is by level.

In the methods section, a brief description is given as to the calculation of the descent rates. The calculation is done following Schenzinger et al. (2017) by finding for each time step the level of zero wind line and computing the height difference of this zero-wind level for consecutive months, i.e., this calculation is not for a single level.

L246, See Geller et al 2016 about GW variations.

Thanks for this suggestion. Our evidence suggests that the GW variations or at the very least the ENSO influence on the QBO through the GW variations is less pronounced in this model which is interesting.

L252, So the frequency would be for example (# months EN) / (# months W)? Maybe mention that IOD will be considered later?

Yes, exactly, we have clarified this in the Table caption and we mention that the IOD will be considered later.

L260, ENSO3.4 -¿ EN3.4 (or maybe ENSO) Done.

Fig 7, [] missing around mm day-1 (check other plots as well). I guess IOD-prc is same as IOD?

Done, the IOD-prc reference has been removed, now we refer to the IOD or the IOD index (based on convective precipitation) consistently.

L266, write months in full. Can you elaborate on how the difference model/obs depends on the ENSO evolution in the model (e.g. Lengaigne et al., 2006)? Also worth nothing how the model index amplitudes are 2-3 times smaller than obs.

The revised manuscript writes months in full.

We agree that the amplitude difference between model/obs indices should be noted in the manuscript (and we have rewritten this section accordingly). However, by boostrapping the long simulation into periods of similar length to the observations, one can find amplitudes in the differences of these indices of similar strength to observations, i.e., certain 36-yr parts of the model simulation agree with the magnitude of the observed differences.

Fig8, as before, why 'convective'? Why now using a higher confidence level?

ERA5 and the simulations output the diagnostic of convective rainfall which is the diagnosed precipitation resulting from the convection scheme-only. The revised manuscript clarifies that we use convective precipitation to investigate more directly a possible link between deep convection and the QBO, for which we use ERA5 and model data. The confidence level written in this figure caption was a typo, as the same confidence level is used throughout the study.

L275, but could this be model-dependent?

Yes, different horizontal resolutions lead to different tropospheric biases in the monsoons which could

change the specific QBO-ENSO impacts for each region. The key message from this figure in our view is that the combined influences appear to be relevant in these models, which means that in some cases part of the non-linearity of ENSO impacts is associated (to a modest extent) with the QBO.

L280, Please avoid the mix of abbreviations and months in full The revised manuscript now uses months in full.

L286, Maybe the Indian Ocean sector, rather than IOD?

Replaced as suggested.

L293, why '.'? Rephrased.

L295, atmospheric circulations. However, the model biases should be noted.

Done. Model biases are now more explicitly mentioned in the discussion section as stratospheric and tropospheric biases need to be carefully considered in all of our results.

L300, How are these longitudes selected?

These longitudes were selected based on the results from the previous sections, i.e., Figures 4 and 5.

Fig9, Only convective, stratiform rainfall removed? Is panel (b) indicating a double ITCZ bias? Can you comment in the text?

Yes, in both cases only precipitation resulting from the convection scheme in the model (convective precipitation) is shown. Panel b) shows a double-ITCZ bias but a true "double" ITCZ is only found for a limited period of time in boreal winter (García-Franco et al., 2020). The revised version now makes this comment.

L317, remove 'rate'

Done.

Fig10, define acronyms MSD, NAM. For more direct comparison you could mask values over oceans? Do you know why the regions show very net boundaries in some cases? Compare with Lee and Wang, 2012 their Fig4

Done. The net boundaries are found by design in the model to look more closely at land monsoon regions and compare with the observational dataset. As this reviewer points out, one alternative would be to mask values over ocean.

Fig11, I am confused by vector sizes. They are 3 or 0.3 10-2 Pa s-1, but their lengths do not differ by a factor 10. Please clarify. Also the plots are quite busy, can you try improving them?

Thank you for pointing out this confusion. This confusion arises because the vector sizes in the climatology plots do not correspond with the vector sizes key in the QBO anomaly plots, this means that the two vector sizes are not related to each other and their lengths are not meant to differ by a factor of 10. We have clarified this confusion in the figure caption. Additionally, we have improved the plot by selecting a smaller vertical extent to more clearly show the differences, and we have also removed the hatching for the significance and instead only shade/color the significant differences and finally, the new plot also uses a slightly modified color scale to make the plots less busy.

L330, Mention the QBO biases which may be important

The revised version of the manuscript now discusses the weak amplitude bias of the QBO in the lower stratosphere which is found in all the CMIP6/QBOi models with varying degrees of magnitude. The HadGEM3 model is amongst the best in this respect, yet the bias is noteworthy still. The mention of the QBO biases is done in several parts of the manuscript including in this section referenced by the reviewer.

L335, If you integrate to the top, then the integration bounds are swapped and  $0 - p_{top}$  (or  $p_{surf}$ )? Gravitational acceleration (g) rather than constant (G)? How do you compute the divergent component of zonal wind?

Thanks for this comment as the text was wrong. As the reviewer mentions, the integration bounds would be swapped. The revised version now states in the text that the methology is to integrate *from* the top level to the surface. We use the acceleration term (g) for consistency with previous studies (see e.g. Yu and Zwiers, 2010). The divergent component of the wind is computed using the python library windspharm (Dawson, 2016); the revised manuscript now makes clarifies this part.

L351, To me some QBO/ENSO superposition can also be seen from the plots.

Agreed, as in most plots a superposition of ENSO impacts can be seen, which is why for all the plots we show a Neutral ENSO version of the figure. In this case, the revised manuscript now notes this remark by the reviewer, i.e., that some superposition can be seen.

L406, or role of QBO bias...

The revised manuscript discusses the weak amplitude bias of the QBO in the lower stratosphere which is found in all the CMIP6/QBOi models with varying degrees of magnitude. The HadGEM3 model is amongst the best in this respect, yet the bias is noteworthy still. The mention of the QBO biases is done in several parts of the manuscript including in this section referenced by the reviewer.

L416, have you ever mentioned TRMM in the text?

We have removed this reference to TRMM.

L420, 'observations' -¿ 'variables'

**Done.**

L422, revise. you speak about days, I understand the input data is monthly mean, so it this weighting already built in? Does the MOHC model have 360 day calendar?

This formulation was written for the most general case, which is the case of the computation of the composites of observational data. However, as the reviewer points out, in the case of the MOHC the calendar is 360-day which means there is no need to consider the "day" part of this formulation.

L435, I guess the 'i' subscript is redundant with one predictor? Same in Fig S3

Yes, in the simplest cases (Figs. S3 a, c) there is only one predictor. The equation has been rewritten and the plot title updated.

L440, State that summation is j = 1...N, as  $X_0$  appears already

Done.

L446, Is there a stray A3?

Typo, the A3 reference has been removed.

L513, why uppercase? Corrected to lower case.

**2. Reviewer 2**

The paper seeks to understand the link between the tropical stratospheric QBO and variability elsewhere in the tropical atmosphere (and on ocean SST). The tools used include observational/reanalysis output since 1979, and several preindustrial control runs from various versions of the Met Office model. This is an interesting subject and the foundation of a paper that could eventually be publishable in WCD is clearly present. Many of the arguments are not convincing in their current form however, and while I don't think these critiques are insurmountable, addressing them will require some major rethinking and rewriting.

We thank this reviewer for their time reviewing our manuscript. We found your concerns and questions to be very useful to revise and improve our submission. Below we present a detailed point-by-point response to your concerns and questions. Your comments are shown in black and our responses in blue.

**2.1. Major comments**

2.1.1. My main criticism is that the authors argue there is a connection between the QBO and ENSO, but the evidence provided is not strong enough (and in this reviewer's opinion the authors' claims are actually incorrect). First, the observational period covered by this paper only begins in 1979, however high-quality radiosondes have tracked the QBO since 1953, and reliable information on the ENSO state is available even earlier. Studies that have used the entirety of the observational record have reached an opposite conclusion of that reached by the authors. Specifically, in the period before 1979, there were more easterly QBO events simultaneous with El Nino. This has been noted by at least three papers (Garfinkel and Hartmann 2007, Hu et al 2012, Domeisen et al 2019), none of which were cited in this paper. The net effect is that the observed connection between ENSO and the QBO is non-stationary, and (cherry-) picking a limited subset of the full observational record (at least until 2018, the last year considered by Domeisen et al 2019), the correlation was essentially zero.

Thank you for this comment. To address this comment we have analyzed the period before 1979 using the zonal winds at 70 hPa from the Freie Universität Berlin (FUB) radiosonde dataset, available at https://www.geo.fu-berlin.de/en/met/ag/strat/produkte/qbo/index.html#access. This dataset covers the 1953-2020 period using radiosondes launched at Canton Island, Gan/Maledive Islands and Singapore. Furthermore, a reconstructed index at 90 hPa, described in Brönnimann et al. (2007), was used to investigate an even longer period (1930-2020), this dataset is available at https://climexp.knmi.nl/data/iqbo\_90.dat. Using these two new indices, the observational part of Figure 3 in the original manuscript was reproduced for different periods, shown in Figure 1 for detrended HadSST data.

Figure 1 confirms the suggestion of this reviewer that in the 1953-2020 period there is little-to-no difference (QBO W-E) in the equatorial Pacific due to the cancellation of a negative signal in 1953-1979 with a positive signal in 1979-2020. However, the longer period (1920-2021) shows large agreement with the shortest 1979-2020 period, except in the South Atlantic. A warmer central-eastern Pacific is observed

---

## Author Response (AR2)

**Manuscript # WCD-2022-14 Response to Reviewers**

García-Franco, JL., Gray LJ., Osprey S., R. Chadwick and Z. Martin

*Atmospheric, Oceanic and Planetary Physics, Department of Physics, University of Oxford. Parks Road Oxford, United Kingdom OX1 3PU email: jorge.garcia-franco@physics.ox.ac.uk*
* * ** * *
**1. Reviewer 1**

The authors provided detailed responses to the points I raised in the previous review stage and explained many aspects of their model configurations.

Since the paper is quite long to read, I would suggest to shorten the text wherever possible (some examples below), otherwise I just have some minor remarks (by line number in the track-changed paper):

We thank this reviewer for their time reviewing our manuscript in this second round of review. We have shortened the manuscript removing the text suggested by this reviewer. Additionally, following the suggestion by Reviewer 2, we have shortened section 3.1 and also removed a couple of sentences in the introduction, all of which amounts to about 15 lines less in the manuscript. Below we present a detailed point-by-point response to your concerns and questions. Your comments are shown in black, our responses in blue and the changes made to the manuscript, where appropriate in violet.

L58-9, since this is also written below, it can be deleted here

Deleted.

L95, daily or monthly?

Monthly, thank you for pointing out this ambiguos statement, we clarified this by changing these lines to the following:

Monthly mean data was used for all observational datasets.

L107, sentence is a bit repetitive

This paragraph was reworded for the (shorter) following text:

The three simulations (GC3 N216-pi, UKESM N96-pi and GC3 N96-pi) used in this study are 500-yr long and have the same experimental design with only 1 ensemble member. The simulations were run with constant year-1850 external forcing, further detail about the MOHC piControl experiments can be found in Menary et al. (2018) and about the UKESM1 model in Sellar et al. (2019).

L122, since this is not the main topic, the sentence is not necessary

Removed.

Fig 3e, how do you combine ensemble members to calculate this time series?

The simulation shown in Fig3e is only GC3 N216-pi which consists of only 1 ensemble member as indicated in section 2.2 and in the paragraph above.

L231, I guess this analysis should be done accounting for underlying trends, so maybe the sentence can be omitted and the topic left for other studies

Rephrased.

L234, 'a multidecadal modulation of...'?

We have removed this text, in order to highlight the advantages of these simulations compared to observations, as suggested by Chaim Garfinkel.

L268, you may want to note that model indices amplitudes are 2-3 times smaller than observations

We have noted so in the text.

L343, NN ENSO -¿ NN

Changed.

L358, extra ')'?

Removed.

L378, for me the 'constant' is this one `https://en.wikipedia.org/wiki/Gravitational_constant`, not 'small g'

We have referred to the 'g' in this context as gravitational acceleration in the revised text.

L459, please rephrase to avoid repetition

Removed.

**2. Reviewer 2**

I thank the authors for their detailed response to my initial comments, including a careful discussion of the differences between their study and that of Rao et al 2020. I appreciate their focus on the role of the QBO level (70hpa vs. 30hPa) and signal to noise ratio (i.e. how many years are included in the composite). I'm going to sign my review, as it is about to become fairly obvious who wrote this review (if it wasn't already obvious from the first round). I also want to apologize if my initial review was a bit too harsh and dismissive, as I really like this paper!

Dear Chaim,

Thank you for your time reviewing our manuscript and your comments which have greatly improved the revised version. No apology was necessary as we understand the reasons for your earlier concerns given the disagreement between our results and those of Rao et al. (2020) and we believe that by investigating these differences and discussing them directly in our manuscript we have improved the paper substantially.

I have one more general comment on section 3.1, and also a few remaining comments on the difference between Rao et al 2020 and this paper.

Section 3.1 still seems to be overly precise when trying to compare the model to observations. The GPCP response shown in figure 1a and the left column of figure 2 reflects the fact that the data is available only from 1979 to the near present. Over this period there were more EN during WQBO. If the observational precip data product was available for more years, then the observed signal would be different (as the authors show shortly for SST). In other words, there is substantial uncertainty on the observed response.

Because of this, I don't think it makes sense in the text to compare the model to observations in a quantitative sense nor to focus on the details of the response, as the observational signal is fundamentally unknown (e.g. Deser et al 2017; Journal of Climate on ENSO teleconnections) and the model SSTs are not the same as obs SSTs.

Stated another way, I would expect the model response to be weaker than obs because the SST response shown in figure 2 is weaker in the ENSO region.

Stated a third way, if we had a gridded, observed precip product for the period 1953 to the near present, I speculate that the agreement with the model would be better.

If the authors agree with my interpretation, the text itself in section 3.1 needs to be modified, though the main conclusions will be generally unchanged (and in fact, the model would actually become more suitable for the analysis the authors subsequently perform).

Thank you for this suggestion. We agree with you that if there was an observational dataset for precipitation available for an earlier period, 1953-2021 or 1921-2021, Figures 1 and 2 would look different. As a result, we have decided to shorten section 3.1 by removing text that compared model and observed responses. However, the Figures themselves set the scene for the rest of the paper so we have not changed

the Figures and we have kept some sentences that indicate that the observational results agree with previous studies. Half-way through section 3.2 we have added the following text:

The observed multidecadal changes to the ENSO-QBO relationship (Fig. 3) means that the precipitation response (Figs. 1 and 2) would likely be different if a longer record of precipitation was available. While our analysis of the observed record is affected by statistical uncertainty (e.g. Deser et al., 2017), this is likely not the case in the pre-industrial control simulations given their length and constant external forcing. This result further highlights the advantage of using these model experiments to understand QBO tropical teleconnections, including ENSO relationships, in the remainder of this paper.

I also have a few comments on the discrepancy between Rao et al and this paper. The first is that Rao et al considered many models where the QBO would be ill-defined at 70hPa. Hence it would be impossible to consider the role of the QBO at 70hPa on impacts outside of the QBO region in such models. While the Met Office models do indeed have a too-weak QBO at 70hPa, this model was actually one of the better ones in this regard (though its periodicity was too long as the authors acknowledge). In order to have a common definition for all models, Rao et al adopted the 30hPa level for all models.

Second, Rao et al identified a tropical convective signal associated with the QBO at 30hPa which differs from the one in this paper in its pattern. Rao et al also identified a robust signal in 100hPa buoyancy frequency for this phase of the QBO in observations and in most models, including the Met Office models which were among the best performing (Figure 9 of Rao et al). My interpretation is not that the winds at 30hPa have a direct effect on buoyancy frequency and convection, rather that this is a convenient way to pick a particular phase of the QBO whose downward extension has a direct impact on the TTL. For this specific phase of the QBO, the Met Office models struggle to represent the convective impact even as they did a reasonable job with the buoyancy frequency anomalies at 100hPa. This could be because of biases in the QBO itself (e.g. downward propagation to the lower stratosphere, or the overly long stalling of lower stratospheric anomalies), or a small signal to noise ratio that a single ensemble member may miss (as the authors point out).

My own speculation/intuition based on the results from Rao et al and the current paper is that there may be multiple QBO regimes with an impact on tropical convection, but future work is clearly needed to sort out whether this indeed the case and why. While I agree that the 70hPa level is best to diagnose a direct impact on the TTL, the unfortunate reality is that nearly all models still struggle with the downward extension of the QBO to the lower stratosphere with very little progress having been made recently and with few ideas on how to improve the situation (other than substantially more resolution, as suggested in Garfinkel et al 2022; JAMES). Hence a focus on 70hPa necessarily excludes many models which may still have teleconnections from the QBO higher up. I would suggest that as a community, we should consider teleconnections associated with different QBO levels (e.g. both 70hPa and 30hPa), so as to be able to include models with relatively larger biases in the QBO in the lowermost stratosphere.

Performing such an analysis is well outside the scope of the authors' paper, and specifically the authors

could decide to not include any of it. However, the authors may want to include more about this sensitivity to QBO level and the nature of biases in most models in their discussion section.

Signed, Chaim Garfinkel

Thank you for these insightful remarks. We agree with your comment that a 70 hPa index would not work for those models in which the weak QBO amplitude bias in the lower stratosphere is so large that the QBO would be ill-defined at this level. The discussion in the manuscript aims to clarify differences between our findings and those of Rao et al. (2020) and not to indicate that only this level should be used for all models. Multi-model analyses should indeed consider multiple levels or an index that captures the state of the vertical profile (e.g. Schenzinger et al., 2017). In the discussion section, in the second-to-last paragraph, we have added the following sentences to highlight the role that stratospheric biases have for diagnosing teleconnections and the uncertainty introduced by the weak amplitude bias in our results.

Tropospheric biases, e.g., in the strength or position of the ITCZ (Fig. 9), may limit the robustness of these results and may mean that the impacts diagnosed in this study may be different in another model. Similarly, stratospheric biases such as the weak amplitude of the QBO in the lower stratosphere found in most models (Bushell et al., 2020, Rao et al., 2020), means that the simulated tropical pathway of QBO teleconnections may be weaker, nonexistent or difficult to diagnose in some models, highlighting the need to improve vertical resolution in GCMs (Garfinkel et al., 2022).

**2.1. Minor comments**

Line 227 please rewrite "for the most part of the simulation"

Reworded to: mostly positive.

Line 242: "However, the equatorial Atlantic and Pacific MAM responses are stronger when ENSO events are included." This isn't obvious to me from figure 4.

We have removed this sentence.

Table 1: I found the caption included for this table confusing. Are the stated units ("#months EN/# months W") correct? Shouldn't it be ("#months ENSO/# months QBO")? Also, "standard deviation of the PDF" is confusing as well – I think you mean you did a bootstrapping in order to quantify the uncertainty of #months ENSO/# months QBO, but maybe I misread.

We have changed the units in the caption as suggesed and yes, a boostrapping of the data was done to account for model and observational uncertainty. This is now clarified in the table caption but also in the text as follows:

Probability density functions (PDFs) were constructed, first, for the observations by bootstrapping with replacement to account for observational uncertainty, and for the model data using 39-yr samples to match the length of the ERA5 period.

Section 3.4 The word "explain" on line 373, 399, and 440 seems overstated. There is no casual explanation here, as the authors note later. Rather the authors are establishing a self-consistent framework or schematic that allows for connecting tropical anomalies in disparate regions.

These sentences have been rephrased.

Deser, Clara, Isla R. Simpson, Karen A. McKinnon, and Adam S. Phillips. "The Northern Hemisphere extratropical atmospheric circulation response to ENSO: How well do we know it and how do we evaluate models accordingly?." Journal of Climate 30, no. 13 (2017): 5059-5082.

Garfinkel, Chaim I., Edwin P. Gerber, Ofer Shamir, Jian Rao, Martin Jucker, Ian White, and Nathan Paldor. "A QBO Cookbook: Sensitivity of the Quasi-Biennial Oscillation to Resolution, Resolved Waves, and Parameterized Gravity Waves." Journal of Advances in Modeling Earth Systems 14, no. 3 (2022): e2021MS002568.

**3. Reviewer 3**

The authors have addressed most of my comments satisfactorily, and I am now ready to accept the paper for publication.

We thank this reviewer for their time reviewing our manuscript a second time and their constructive comments. Below we present a detailed point-by-point response to your concerns and questions. Your comments are shown in black,our responses in blue and the changes made to the manuscript, where appropiate in violet.

However, I am still confused about the different significance tests. For the composite analysis you use bootstrapping for the observations but a Welch t-test for the models? This is what you now write in the paper, but from the Reply you seem to also use the t-test for observations? Why not use the same test for both observations and models?

The main reason is because the same test would not be well suited for both observations and models. Note the very different sample sizes of the observations and the long simulations. The composite sizes of a difference such as QBO W-E (NN) (Fig 11c) would be too small (N 10 or less) for anything meaningful to be drawn from a t-test. However, the bootstrap with replacement test as done in the manuscript allows to evaluate the likelihood of a difference being affected by observational uncertainty. We have also used a bootstrap test in the model simulations for most of our results, but this bootstrapping is different by design. In this case we compute the QBO W-E difference for model samples of randomly sampled 39-yr periods, i.e., subsampling bootstrapping, repeated 10,000 times, and we evaluated how likely it is that our result is of the same sign as the mean QBO W-E difference. In other words, this process is very different to the observed bootstrapping, repeating the same process for the model would mean boostrapping with replacement in a composite with such a large size that this test would not render meaningful results. Our investigation shows that the modelling results discussed in the manuscript are not sensitive to the choice of test (Welch-test versus boostrapping into 39-yr chunks). The revised text in this section now reads:

The significance level is then interpreted as QBO W-E differences that are outside of the 95% of the distribution of randomly generated differences. The significance of the differences in the simulations is estimated using a Welch two-sided t-test, but other bootstrap methods were tested without significantly changing the results.

When considering correlations (and regressions?) a bootstrap method is also used. But I would think serial correlations should be taken into account. You can do this by using block-bootstrap.

Thank you for pointing this out, the text was misleading, the reported significance in Figure 5 uses the standard p-values from the regression model, which are based on the $t$ distribution and the standard error. Thank you for the suggestion of using a btlock-bootstrap for our regression analyses. While the QBO index is definitely autocorrelated, the deseasonalized precipitation time-series at each grid point are most frequently not autocorrelated. We have computed the regression coefficients and the p-values using block bootstrapping as follows. Circular block bootstraping was used following (Politis and Romano,

[Figure]

Figure 1: Regression coefficients between the QBO index and deseasonalized convective precipitation (a) as in Figure 5 of the main manuscript, (b) as (a) but showing the mean regression coefficients from a circular block-bootstrap method (see text) using a block length of 61 months.

1991) using block lengths of 21, 61 and 121 months and using 10,000 repetitions. For each repetition, the regression coefficient and the p-value are computed and stored, then, the mean values are plotted for the block length of 61 months (Figure 1). The original results hold, although several sparse regions do show a change in the magnitude of the regression coefficient and overall less significant regions are diagnosed in block-bootstrapping results. The main possible reason for these results is the lack of autocorrelation in the deseasonalized precipitation which means the regression analysis renders effectively equal results using block bootstrap methods or straightforward regression analyses. The figure caption (Fig. 5) in the manuscript now specifies:

"..and the hatching indicates significance to the 95% confidence level based on a t-test."

In Equation A1 the subscript i should be deleted.

Done.

**References**

Bushell, A. C., Anstey, J. A., Butchart, N., Kawatani, Y., Osprey, S. M., Richter, J. H., Serva, F., Braesicke, P., Cagnazzo, C., Chen, C.-C., Chun, H.-Y., Garcia, R. R., Gray, L. J., Hamilton, K., Kerzenmacher, T., Kim, Y.-H., Lott, F., McLandress, C., Naoe, H., Scinocca, J., Smith, A. K., Stockdale, T. N., Versick, S., Watanabe, S., Yoshida, K. and Yukimoto, S. (2020), 'Evaluation of the Quasi-Biennial Oscillation in global climate models for the SPARC QBO-initiative', *Quarterly Journal of the Royal Meteorological Society* pp. 1–31.
   **URL:** *https://rmets.onlinelibrary.wiley.com/doi/abs/10.1002/qj.3765*

Deser, C., Simpson, I. R., McKinnon, K. A. and Phillips, A. S. (2017), 'The northern hemisphere extratropical atmospheric circulation response to enso: How well do we know it and how do we evaluate models accordingly?', *Journal of Climate* **30**(13), 5059–5082.

Garfinkel, C. I., Gerber, E. P., Shamir, O., Rao, J., Jucker, M., White, I. and Paldor, N. (2022), 'A qbo cookbook: Sensitivity of the quasi-biennial oscillation to resolution, resolved waves, and parameterized gravity waves', *Journal of Advances in Modeling Earth Systems* **14**(3), e2021MS002568.

Menary, M. B., Kuhlbrodt, T., Ridley, J., Andrews, M. B., Dimdore-Miles, O. B., Deshayes, J., Eade, R., Gray, L., Ineson, S., Mignot, J., Roberts, C. D., Robson, J., Wood, R. A. and Xavier, P. (2018), 'Preindustrial control simulations with HadGEM3-GC3. 1 for CMIP6', *Journal of Advances in Modeling Earth Systems* **10**(12), 3049–3075.

Politis, D. N. and Romano, J. P. (1991), *A circular block-resampling procedure for stationary data*, Purdue University. Department of Statistics.

Rao, J., Garfinkel, C. I. and White, I. P. (2020), 'How does the quasi-biennial oscillation affect the boreal winter tropospheric circulation in cmip5/6 models?', *Journal of Climate* **33**(20), 8975–8996.

Schenzinger, V., Osprey, S., Gray, L. and Butchart, N. (2017), 'Defining metrics of the Quasi-Biennial oscillation in global climate models', *Geoscientific Model Development* **10**(6).

Sellar, A. A., Jones, C. G., Mulcahy, J., Tang, Y., Yool, A., Wiltshire, A., O'Connor, F. M., Stringer, M., Hill, R., Palmieri, J., Woodward, S., de Mora, L., Kuhlbrodt, T., Rumbold, S., Kelley, D. I., Ellis, R., Johnson, C. E., Walton, J., Abraham, N. L., Andrews, M. B., Andrews, T., Archibald, A. T., Berthou, S., Burke, E., Blockley, E., Carslaw, K., Dalvi, M., Edwards, J., Folberth, G. A., Gedney, N., Griffiths, P. T., Harper, A. B., Hendry, M. A., Hewitt, A. J., Johnson, B., Jones, A., Jones, C. D., Keeble, J., Liddicoat, S., Morgenstern, O., Parker, R. J., Predoi, V., Robertson, E., Siahaan, A., Smith, R. S., Swaminathan, R., Woodhouse, M. T., Zeng, G. and Zerroukat, M. (2019), 'UKESM1: Description and evaluation of the UK Earth System Model', *Journal of Advances in Modeling Earth Systems* **11**(12), 4513–4558.